∂ | **Open Peer Review** | Host-Microbial Interactions | Methods and Protocols

# Bridging the gap: organotypic models to study late-onset group B streptococcus infection

Alexia N. Pearah,[1] Nichol John-Lewis Edwards,[2] Rico R. Carter,[1] Leigh-Anne M. Worthington,[1] Madison R. Huszar,[1] Kristen Domínguez,[2] Hannah L. Wapshott-Stehli,[1] April K. Lindon,[1] Sophie E. Darch,[2] Thao T. B. Ho,[1,2] Tara M. Randis[1,2]

**ABSTRACT** Group B *Streptococcus* (GBS) is a major cause of neonatal sepsis worldwide. Gastrointestinal colonization by GBS is an important risk factor for late-onset disease in newborns. Most studies of GBS pathogenesis rely on animal models with limited human relevance or use immortalized adult human cells that do not recapitulate the unique genetic and phenotypic features of the neonatal intestinal epithelium. Previous studies using tissue-derived human intestinal enteroids (HIEs) and induced pluripotent stem cell-derived human intestinal organoids (HIOs) to model host interactions with enteric pathogens have yielded valuable insights. Here, we describe the use of GBS-exposed HIEs and HIOs to study GBS interactions with the immature human intestinal epithelium. Using these models, we demonstrated that GBS induces changes in gene expression of both HIEs and HIOs that are distinct from what has been reported in immortalized adult cell lines. We observed GBS attachment to the apical surfaces of HIEs and HIOs and, in some cases, translocation across intestinal epithelial barriers. To examine the impact of GBS exposure on intestinal barrier function, we generated polarized HIE monolayers on Transwell plates. We observed GBS translocation across monolayers, accompanied by a trend in increased epithelial barrier permeability reflected by decreased transepithelial electrical resistance. These data demonstrate that both HIEs and HIOs are robust and useful models for studying the pathogenesis of late-onset GBS infection in the vulnerable newborn host. Importantly, they provide a much-needed platform to test novel preventative strategies.

**IMPORTANCE** Group B *Streptococcus* (GBS) is a major cause of infectious morbidity and mortality in neonates. Late-onset GBS disease, which develops after the first week of life, arises when GBS colonizes the neonatal gut and compromises intestinal barriers, resulting in systemic infection. Studies investigating the pathogenesis of late-onset GBS disease typically employ animal models or *in vitro* immortalized adult human intestinal cell lines, which can limit the applicability of findings to human neonates. In this study, we introduce the use of three-dimensional fetal tissue-derived human intestinal enteroids and induced pluripotent stem cell-derived human intestinal organoids to study GBS-host interactions within the gut. These novel models demonstrate improved ability to recapitulate the vulnerability of the immature human host and function as a platform to test novel interventional strategies to protect exposed newborns.

**KEYWORDS** group B streptococcus, *Streptococcus agalactiae*, sepsis, organoid, enteroid

G roup B *Streptococcus* (GBS) is a leading cause of neonatal sepsis globally, and colonization of the newborn gastrointestinal tract is a critical first step in the pathogenesis of late-onset disease. Studies of late-onset disease pathogenesis typically utilize animal models with limited human relevance or *in vitro* adult human intestinal cell lines that are a popular alternative to studying gastrointestinal disease but do not closely

**Peer Reviewer** Raghavendra Nagampalli, St. Jude Children's Research Hospital, Memphis, Tennessee, USA

Address correspondence to Alexia N. Pearah, alexiapearah@usf.edu.

The authors declare no conflict of interest.

See the funding table on p. 13.

recapitulate the unique vulnerabilities of the neonatal host, such as reduced mucosal barrier function and distinct immune signaling cascades. Furthermore, immortalized adult cell lines do not differentiate into intestinal epithelial cell (IEC) subtypes (goblet cells, Paneth cells, etc.) that have been shown to play a key role in host defense. Modeling GBS intestinal colonization in neonatal/juvenile animals enables the study of complex interactions between GBS and the immature mucosal barrier; however, the translation of results to the human host remains limited (1–3).

Human intestinal enteroids (HIEs) derived from fetal tissue offer a major advantage in that they retain the unique cellular programming of the immature host (4, 5). The intestinal crypt cells from fetal tissues can differentiate into enteroids comprised of enterocyte subtypes, including microfold (M) cells, goblet cells, and Paneth cells. Similarly, HIOs, derived from induced pluripotent stem cells, recapitulate key features of fetal intestinal development and include multiple enterocyte subtypes. They also have the added advantage of differentiating into mesenchymal populations such as fibroblasts and myofibroblasts. This multicellular composition provides a physiologically relevant model of the intestinal niche, enabling the study of epithelial-mesenchymal interactions during development and disease.

The luminal surface of enterocytes, which is enclosed within the interior of HIE and HIO spheroids, is difficult to access. To overcome this challenge, polarity reversal of HIEs/HIOs is generated to produce apical-out HIEs and HIOs, allowing for the luminal apical surface of IECs to be in direct contact with reagents and/or microbes added to culture media (6). These three-dimensional (3D) structures can also be transformed into polarized monolayers to examine the functional permeability of IECs using Transwell systems.

Previous research utilizing adult-host-derived intestinal enteroids has yielded significant insights into the mechanisms underlying infections by various enteric pathogens, including *Shigella* (7), *Salmonella* (8), *Escherichia coli* (9, 10), noroviruses (11–13), rotavirus (14, 15), and *Clostridioides difficile* (16). Immature human intestinal enteroids (HIEs) have been specifically employed to model the neonatal gut in studies investigating the pathogenesis of necrotizing enterocolitis (4) and *E. coli* infection (17). Similarly, HIO-based models have facilitated the study of enteric pathogens (18, 19). The use of such organotypic models has yet to be used in studies of GBS pathogenesis. This study establishes models utilizing GBS-exposed apical-out HIEs and HIOs, as well as polarized HIE monolayers, to examine GBS-host interactions.

## RESULTS

### GBS adheres to and invades the apical surface of both HIEs and HIOs

Apical-out orientation of HIOs and HIEs was confirmed by using a villin-specific stain to identify the epithelial brush border due to villin being a protein that is only expressed on the apical side of the intestinal epithelium, as well as staining for F-actin along with nuclei (Fig. S1). Dampened signaling of villin within basolateral-out HIE, compared to the prominent signaling of villin in apical-out HIE, further confirmed that polarity reversal was achieved (Fig. S1A and B). F-actin and nuclei staining of basolateral-out HIE revealed F-actin internal to the nuclei (Fig. S1C), whereas in apical-out HIE, F-actin revealed distinct prominence externally to the nuclei (Fig. S1D). Staining of villin with nuclei was completed for both basolateral-out HIO (Fig. S1E) and apical-out HIO (Fig. S1F), verifying that polarity reversal was accomplished similarly to HIE. F-actin and nuclei staining were also completed for basolateral-out and apical-out HIOs (Fig. S1G and H). iPSC-derived HIOs contain both epithelial and mesenchymal cell populations, resulting in increased structural complexity. Because F-actin is expressed in multiple cell types, including mesenchymal cells, actin staining in HIOs does not selectively mark the apical brush border and therefore cannot reliably indicate polarity orientation. For actin staining to accurately indicate polarity in HIOs, the mesenchymal cells must be removed, resulting in a model composed primarily of epithelial cells, such as enteroids. Differentiation into distinct HIE cell types was confirmed previously in the basolateral

orientation (5) and in the apical-out configuration (Fig. S2). HIO cell types were validated by completing qRT-PCR (Fig. S3). Figure 1A illustrates the generation of HIE/HIO, and Fig. 1B depicts the subsequent immunostaining procedure. Co-incubation of GBS with HIEs for 2 h results in GBS adherence to the apical surface of IECs (Fig. 2A). In some cases, GBS breaches the apical surface of HIEs and can be visualized within the enteroid (Fig. 2B through G). To allow for enhanced visualization of internalized GBS, the QR code video of Fig. 2D displays the nuclei and GBS rendered with raw filter of villin, highlighting GBS within the enteroid. Similarly, following co-incubation of GBS with apical-out HIOs, GBS is visualized adhering to the apical surface of IECs (Fig. 3A through D) and, in instances, is visualized within the organoid (Fig. 3G).

## GBS induces distinct patterns of gene expression in HIEs and HIOs

Previous studies by our group using both a mouse model of GBS intestinal colonization and Caco-2 cell lines revealed GBS-induced alterations in IEC gene expression—specifically in those related to barrier function (21). We moved to validate these findings using both HIE and HIO models. In HIEs, we noted a significant increase in expression of the gene encoding the protein claudin-2 (*CLDN2*) and decreased expression of nitric oxide synthase 2 (*NOS2*) and olfactomedin 4 (*OLFM4*) genes when compared to the vehicle control (Fig. 4A). These alterations in *CLDN2* and *NOS2* expression aligned with our previous study using small intestinal mouse tissue (21). In contrast, when we exposed HIOs to GBS, we observed a significant decrease in *CLDN2* expression and no significant changes in *NOS2* and *OLFM4* expression (Fig. 4B). In Caco-2 cells, we noted again an increase in *CLDN2* expression and decrease in *OLFM4* expression following GBS exposure, similar to that observed in HIEs, and an increase in *NOS2* expression (Fig. 4C).

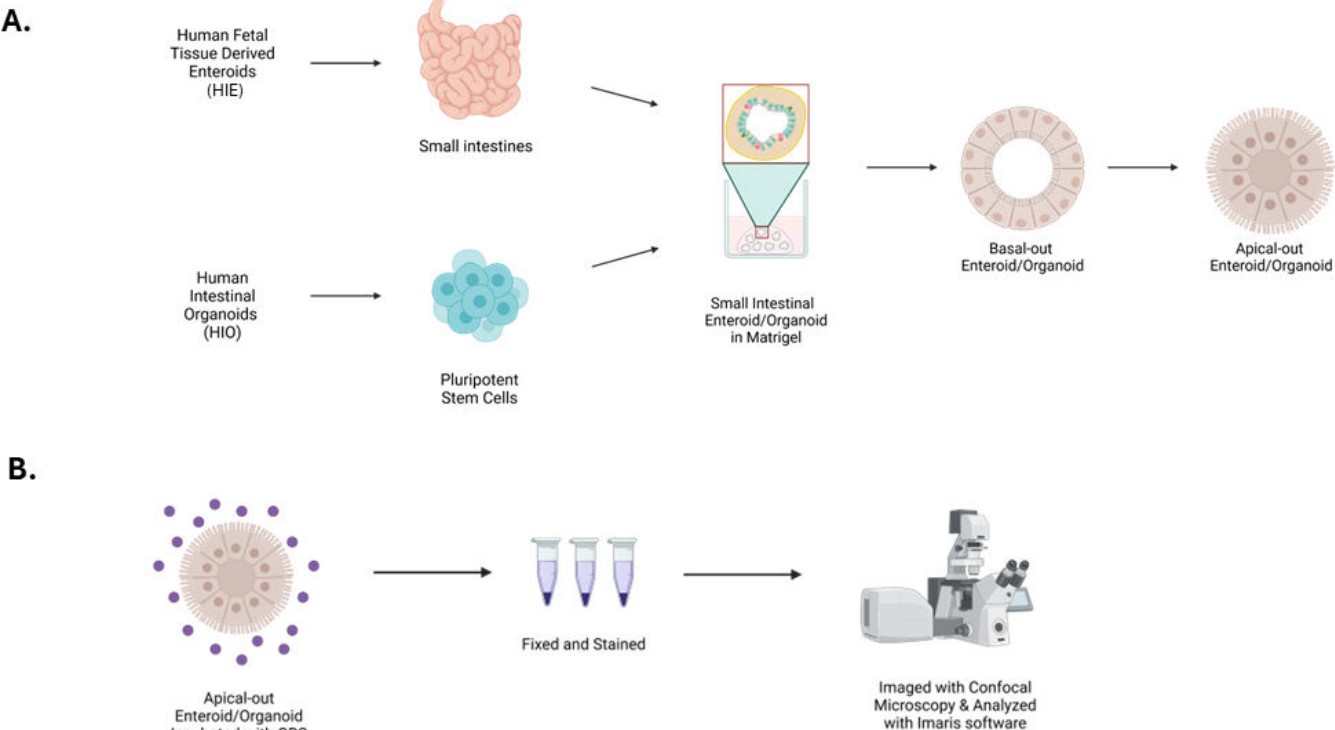

**FIG 1** Graphical illustration methods. (A) The generation of apical-out enteroids/organoids, and (B) the infection, staining, and imaging protocol. This image was created using Biorender.

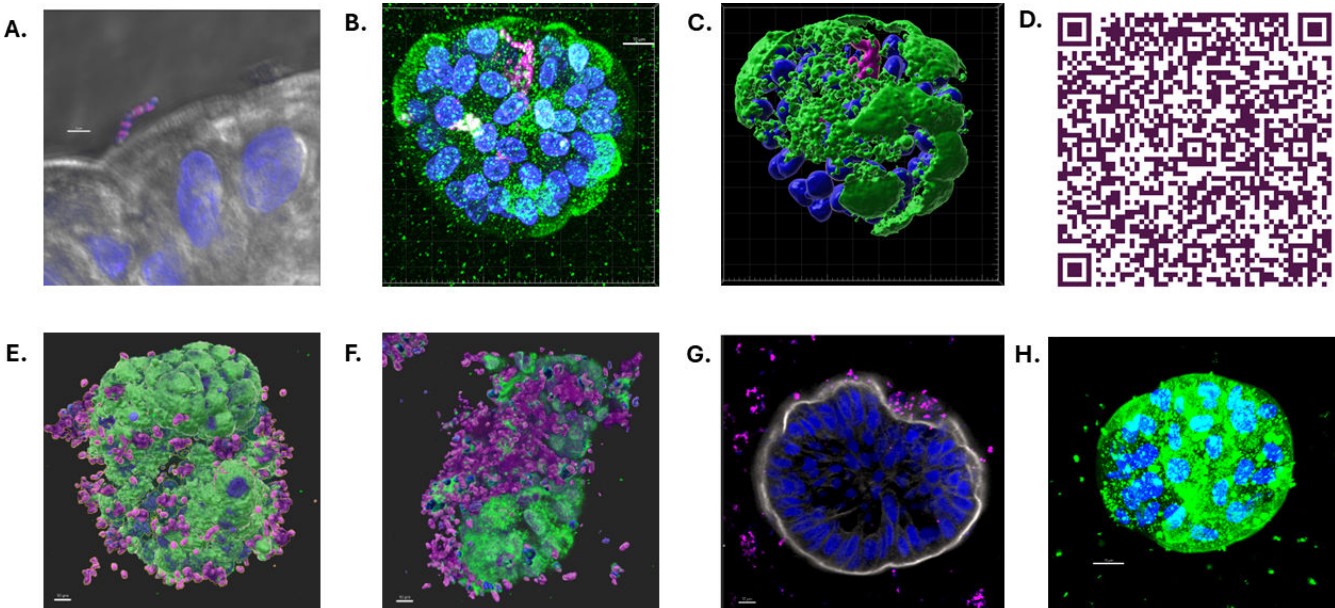

**FIG 2** Laser confocal scanning microscopy imaging of GBS-exposed HIE. (Blue = nuclei, DAPI; green = villin, anti-villin IgG Alexa Fluor 488; magenta = GBS, anti-GBS IgG Alexa Fluor 568; and white = F-actin, Phalloidin Alexa Fluor 488.) (A) Increased magnification of GBS adhering to the epithelial cell surface. (B) GBS is visualized within the apical-out enteroid. (C) Three-dimensional rendering of image D. (D) QR code link to 3D Vimeo video (20). The video displays nuclei and GBS rendered with a raw filter of villin, allowing enhanced visualization of internalized GBS. (E and F) GBS adhesion and invasion of HIE enteroid using 3D rendering of confocal images. (G) GBS-infected enteroid with nuclei and F-actin staining. (H) Representative image of HIE control (no GBS exposure) shown for contextual comparison. Scale bars: Panel A, 3 µm; Panel B–G, 10 µm; and Panel H, 15 µm.

## Enteroid monolayers reveal the impact of GBS on epithelial membrane permeability

HIEs can be manipulated to form polarized monolayers with intact tight junctions between IECs, as reflected by increasing transepithelial electrical resistance (TEER) measurements over time. Upon GBS challenge, we noted a trend in increased monolayer permeability as compared to unexposed HIEs, as indicated by a change in TEER. We included EGTA, a known disruptor of adherens and tight junctions, and Triton X-100 as controls. Similar to GBS, EGTA induced a trend in decreased permeability, while Triton X-100 treatment led to complete disruption of barrier function (Fig. 5A). A lactate dehydrogenase (LDH) cytotoxicity assay was performed on media collected from each of the wells. Neither GBS nor EGTA treatments were associated with IEC cytotoxicity, while Triton X-100 treatment induced cell death (Fig. 5B). Recovery of GBS from the media in the basolateral chamber indicates bacterial translocation through the epithelial monolayer (Fig. 5C). These combined data suggest that translocation of GBS across HIE monolayers occurs without inducing cell death via either transcellular or paracellular pathways. A representative confocal microscopy image demonstrated GBS adherence to the HIE monolayer following infection (Fig. 5D).

## DISCUSSION

Investigations into the pathogenesis of late-onset GBS have been impeded by the lack of adequate models to recapitulate the unique vulnerability of the human neonatal host. Here, we describe the use of HIEs and HIOs to validate previous data generated using animal models and immortalized adult cell lines and as a robust tool to study the interactions of GBS with immature enterocytes. The ability to manipulate enteroid and organoid polarity allows for direct visualization of GBS adherence and invasion of luminal surfaces. Polarized monolayers enable the examination of the impact of GBS-induced

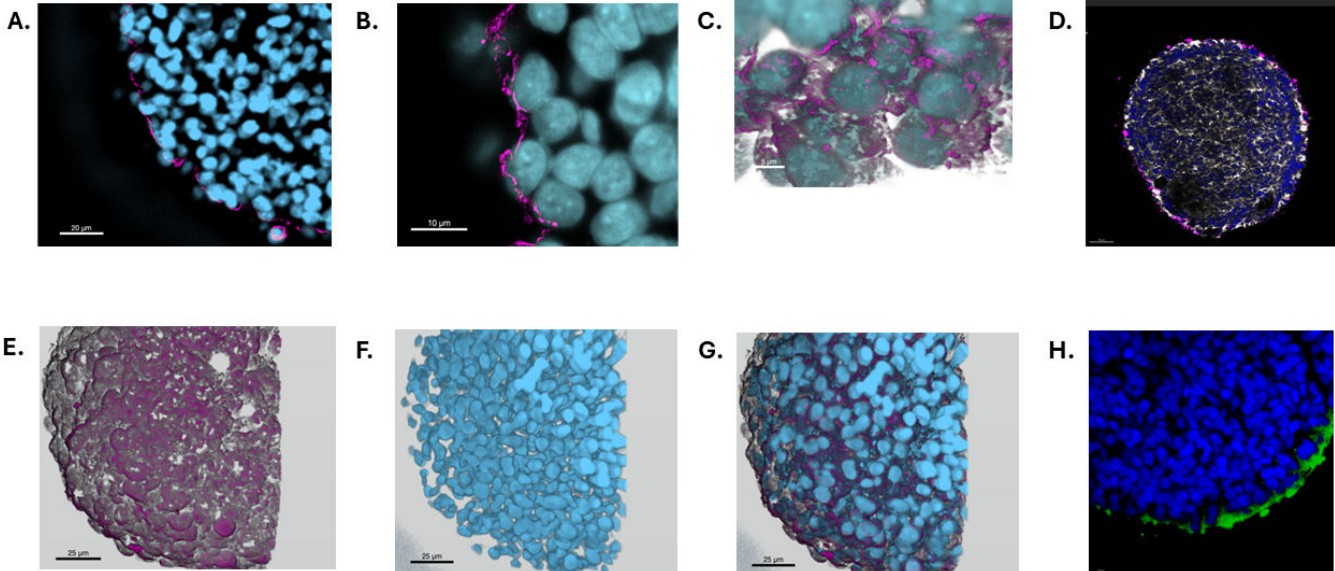

**FIG 3** GBS adherence to the apical-out epithelial border of human intestinal organoids. Representative images using laser confocal scanning microscopy at (A) 63× magnification, (B) 1.5× zoomed in at 63× magnification; thus, the image is at 94.5× magnification, and (C) 100× magnification. Three-dimensional rendering of organoid architecture from confocal z-stack images acquired using a Zeiss LSM880 microscope equipped with a 63× oil objective. Optical sections were collected at 0.5 µm intervals and rendered in Imaris (v10.2, Oxford Instruments) using the *Surface* tool with fixed intensity thresholds applied to each fluorescence channel (SD1). (D) GBS (magenta = anti-GBS IgG Alexa Fluor 568)-infected organoid with nuclei (blue = DAPI) and F-actin staining (White = F-actin, Phalloidin Alexa Fluor 488) (E) of GBS (magenta = anti-GBS IgG Alexa Fluor 568), (F) nuclei (blue = DAPI), and (G) composite image. (H) Uninfected HIO control image stained for nuclei (blue = DAPI) and villin (green = anti-mouse IgG Alexa Fluor 488) included for reference alongside GBS-infected HIO. Scale bars: Panel A, 20 µm; Panel B, 10 µm; Panel C, 5 µm; Panel D, 50 µm; E–G, 25 µm; and Panel H, 10 µm.

disruption of intercellular junctions and epithelial permeability (6, 8–10), culminating in its translocation across IEC barriers.

Previous studies by our group using both a mouse model of late-onset disease GBS intestinal colonization as well as Caco-2 cell lines revealed GBS-induced alterations in IEC gene expression—specifically in those related to barrier function. We conducted a transcriptomic analysis of the mouse model that revealed changes in the KEGG pathway responsible for epithelial barrier structure and function and validated those

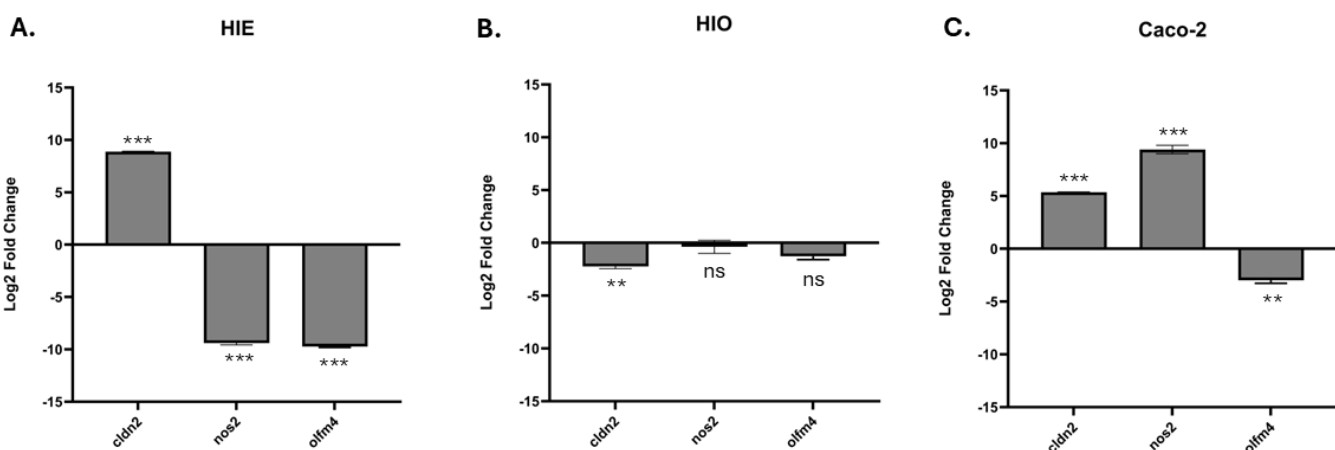

**FIG 4** GBS-induced changes in IEC gene expression. Relative gene expression of *CLDN2, NOS2,* and *OLFM4* normalized to *GAPDH* expression in apical-out HIE (A) and HIO (B) and normalized to *β-actin* expression in GBS-exposed Caco-2 cells (C). The untreated control condition was used as the baseline reference, with its expression level normalized to 1.0 for $\log_2$ fold change calculations using the ΔΔ-CT method. All plotted data represent the $\log_2$ fold changes in expression of each target gene relative to this baseline (unpaired *t*-test, ***$P < 0.0001$, **$P < 0.004$, ns = nonsignificant).

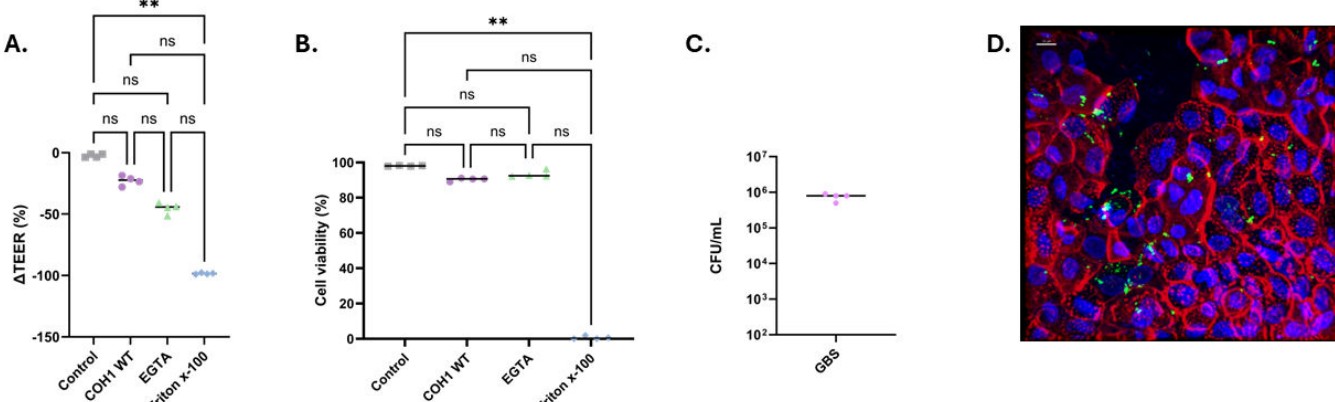

**FIG 5** GBS-induced changes in membrane permeability. (A) TEER measurements in HIEs following 2-h incubation in the following conditions: no infection/treatment control, GBS (COH1 WT) exposed, EGTA-positive control, and Triton X-100-positive control. The results were displayed as a percent change in the TEER reading before and after 2-h incubation (Kruskal-Wallis with Dunn's correction, **$P < 0.001$, ns = nonsignificant). (B) LDH cytotoxicity assay (Kruskal-Wallis with Dunn's correction, **$P < 0.001$, ns = nonsignificant) following 2-h GBS exposure displayed as percent cell viability. (C) Translocation assay (reported as CFU/mL, limit of detection is $10^2$). (D) Representative confocal image of GBS-infected HIE monolayer plate (blue = nuclei, DAPI; green = GBS, anti-GBS IgG Alexa Fluor 568; red = F-actin, Phalloidin Alexa Fluor 488). Scale bar: 10 µm.

findings with Caco-2 cells through targeted qRT-PCR (21). This study prioritized the selected genes based on importance to epithelial barrier function upon GBS exposure across the models of Caco-2 cells, HIEs, and HIOs. We observed key differences in host gene responses between the three models (*CLDN2, NOS2,* and *OLFM4*). This aligns with previous publications reporting differences in host response between intestinal organotypic models and immortalized cell lines such as Caco-2 and T84 cells following infection with norovirus, *E. coli*, and *C. difficile* (13, 16, 22). Of note, we chose different housekeeping genes for this analysis: *β-actin* was used in Caco-2 cells and *GAPDH* in HIE and HIO models. This was based on literature (15, 23–26), as well as validation of the stability of these housekeeping genes performed in our own lab. Overall, these results highlight the importance of HIE and HIO as experimental models to validate and complement existing models of GBS-host interactions.

The reversed polarity of apical-out HIEs and HIOs allows easy access to the lumen by simply adding GBS to the culture media. This eliminates the need for technically demanding microinjection through the basolateral membrane, which results in barrier disruption. However, HIE and HIO spheroids may vary in size and cell number, posing a challenge to standardize the pathogen exposure using typical multiplicity of infection (MOI) ratios.

Similar to other model systems, HIEs and HIOs have specific considerations. Enteroids generated from fetal intestinal tissues reflect the intestinal epithelium of preterm infants (27)—a population most at risk for late-onset GBS infection. While this offers relevant insight, these models may not fully capture the epithelial responses of term infants. Variations in enteroid source tissues and culture conditions can influence reproducibility (28). Additionally, current HIE and HIO systems do not incorporate certain physiological features of the intestinal microenvironment—such as mechanical forces like peristalsis—that may shape GBS-IEC interactions. These models also lack components like the neurovasculature, immune cells, and resident microbiota, which are integral to the *in vivo* intestinal mucosa. Despite these limitations, HIEs and HIOs remain valuable tools that continue to evolve toward more physiologically integrated systems.

Future studies using these models will enable identification of specific IEC subtypes (Paneth cells, goblet cells, enteroendocrine cells, and M cells) involved in GBS adhesion and invasion (3, 29). Furthermore, co-culture of enteroids and organoids with host immune cells, such as macrophages and neutrophils, may be employed to examine interactions among GBS, IECs, and innate immune cells (30). These multicellular model

systems may recapitulate complex intracellular signaling underlying the host response to GBS. Importantly, because the maturity of these organotypic models may be manipulated (22, 31), there is an opportunity to directly compare the response of the adult human host with that of the neonate.

In conclusion, the use of organotypic models to study the pathogenesis of late-onset GBS in the immature human can provide key insights into the progression from host colonization to invasive infection. These models complement existing animal models and may accelerate the development of novel therapeutic or preventative strategies.

## MATERIALS AND METHODS

### Cultivation and passaging of HIEs

Human intestinal enteroids (HIEs) were cultivated from normal, de-identified human fetal small intestinal tissue obtained from the NICHD-supported Birth Defects Research Laboratory (BDRL, R24HD000836) at the University of Washington. The small intestine specimens were collected, stored in ice-cold Dulbecco's phosphate-buffered saline (DPBS), and mailed overnight, followed by processing as previously described (5). HIEs were cultivated, and monolayers were generated following previously established protocols from the Tissue Modeling Laboratory at the University of Michigan (32) with minor adjustments and adaptations from a previous study by Lerena et al. (5), Tsai et al. (33), and Jung et al. (34). Upon receiving a small intestine specimen, the tissue was washed with cold DPBS to remove fecal matter and cut into approximately 5 mm wide sections. Samples were incubated for 20 min in cold DPBS with 2.5 µg/mL amphotericin on a Petri dish. Tissue was placed into a new Petri dish with approximately 1 mL of cold enteroid growth medium (100% IntestiCult organoid media from STEMCELL Technologies with 2.5 µM CHIR99021, 10 µM Y27632, and 100 µg/mL Primocin) on ice, and the epithelial cells were scraped with forceps from the fascia. The cells were transferred to 285 µL of the basement membrane protein mix. The basement membrane matrix used was the Cultrex UltiMatrix Reduced Growth Factor Basement Membrane Extract (R&D Systems) and was diluted to a final concentration of 2.5 mg/mL. A recipe for the basement membrane protein can be found in a study by Llerena et al., 2023 (5). Cells with basement membrane matrix mix were gently triturated on ice using a cut, BSA-coated 200 µL tip 20×. Five streaks of 50 µL each of the cell basement membrane matrix mix were placed into one well of a prewarmed six-well plate and incubated for 15 min at 37°C with 5% $CO_2$; 2 mL of the prewarmed enteroid growth medium was added following incubation. The medium was changed daily, and after 10–14 days, HIE were passaged into one well of a six-well plate according to the previously established protocol with adjustments (32).

For HIE passages, the protocol was adapted from a previous study (35) and Michigan Protocols (32, 35). Enteroids were passaged according to cell density to determine the ratio needed for passage, ranging from 1:2 to 1:4 every 4–6 days. The plate was placed on ice, the media were aspirated, and 1.5 mL of cold human colonoid media (HCM) was added to the well. The HCM recipe can be found in Michigan Protocols and is adapted from previous studies (32, 34, 36–38). A cell scraper was vigorously used to detach enteroids embedded in basement membrane matrix from the plate and transferred into a BSA-coated 15 mL conical tube. Cells were triturated 60× with a 0.1% BSA-coated 200 µL tip and centrifuged at 300 × $g$ for 3 min at 4°C. The supernatant was removed, careful not to disturb the cell pellet, and resuspended in the basement membrane matrix mix; 10 µL domes of HIE embedded in the basement membrane were placed into the prewarmed six-well plate. The enteroids were maintained in enteroid growth media changed every other day, comprised of 85% HCM and 15% Intesticult growth media supplemented with 10 µM-Y27632, 100 µg/mL primocin, and 2.5 µM CHIR99021. Surrounding wells without HIE contained 2 mL of DPBS for humidification.

## HIO generation and passaging

Human intestinal organoids were generated using the previously established protocol from Stem Cell Technologies with minor adaptations (39) using the STEMdiff Intestinal Organoid Kit (Cat. no. 05140); 16 µL of basement membrane (Matrigel hESC –Qualified Matrix (Corning) diluted to a concentration of 2%) was combined with 1 mL of cold Dulbecco's modified Eagle's medium (DMEM)/F12, and 1 mL of the mixture was used to coat each well of the six-well plate. The plate was incubated for 1 h at 37°C with 5% $CO_2$. Cell line used was Healthy Control Human iPSC Line, Female, SCTi003-A (Stem Cell Technologies cat. no. 200-0511) and was thawed in 37°C water bath with gentle shaking until mostly thawed with small frozen pellet remaining. Thawed cells were added to 7 mL of mTeSR Plus media (40) and centrifuged at $300 \times g$ for 5 min at room temperature. The medium was aspirated, and the cells were resuspended in 1 mL of mTeSR Plus media. Basement membrane matrix was aspirated from the well of the previously coated plate, and 2 mL of mTeSR Plus medium was added, followed by 1 mL of cell suspension, for a total of 3 mL. Plated cells were placed in the incubator at 37°C with 5% $CO_2$, and the media were changed every other day for approximately 7 days until confluent. After confluency was reached, the cells were expanded to seed a six-well plate; 1 mL of ReLeSR was added to the well, immediately aspirated, and incubated at room temperature for about 4 min, followed by inactivation by adding 1 mL of mTeSR Plus media to allow for cell detachment from the plate. In total, 20 µL of cells were added to each well of a newly coated six-well plate containing 2 mL of mTeSR Plus media and placed in the incubator at 37°C with 5% $CO_2$ for approximately 7 days until confluent.

Once confluency was reached, clump passaging was completed by first coating a 24-well plate as previously described using the same method as the six-well plate with the basement membrane matrix. Adaptation to Stem Cell Technologies passaging protocol was that mTeSR Plus media were always used in place of using mTeSR 1 media. The full six-well plate was removed from the incubator, the media were aspirated, and 1 mL of Gentle Cell Dissociation Reagent was added to each well. Cells were incubated at room temperature for 8 min to allow for cells to clump, and then, the reagent was aspirated; 1 mL of mTeSR Plus was added to each well, and cell aggregates were transferred to a 50 mL conical tube. Cell clumps were counted (~50–200 µm in size) and were seeded as 6,000 clumps in 500 µL of mTeSR Plus media per well of the pre-coated 24-well plate. The plate was incubated at 37°C with 5% $CO_2$ until confluent, at which point the differentiation process was started.

The differentiation process is comprised of three stages: stage 1, definitive endoderm; stage 2, mid-/hindgut; and stage 3, organoid formation. Specialized media that are stage-specific to produce HIOs from induced pluripotent stem cells (iPSCs) are used. Stage 1 began on Day 0, with preparation of the DE Medium (STEMdiff Endoderm Basal Medium + STEMdiff Definitive Endoderm Supplement CJ) as described in the Stem Cell Technologies protocol (39) and warmed to 37°C. Media were aspirated from the confluent iPSCs, and 700 µL of DE medium was added to each well and incubated at 37°C with 5% $CO_2$ overnight. On Day 1, media were aspirated, and 500 µL of DE medium was added to each well and incubated at 37°C with 5% $CO_2$ overnight. The process was repeated for Day 2. Stage 2 began on Day 3 to start differentiation to mid/hindgut, and MH medium (STEMdiff Endoderm Basal Medium + STEMdiff Gastrointestinal Supplement PK + STEMdiff Gastrointestinal Supplement UB) was prepared as described in the Stem Cell Technologies protocol and warmed to room temperature (15°C –25°C). Media were aspirated, replaced with 500 µL of MH medium, and incubated at 37°C with 5% $CO_2$ until the next day. From Days 4 to 9, the medium was changed daily with MH medium. On Day 10, stage 3 organoid formation was initiated by completing the following steps: intestinal organoid growth medium (OGM) was prepared as previously described in STEMCELL Technologies protocol by combining STEMdiff Intestinal Organoid Basal Medium and STEMdiff Intestinal Organoid Supplement. Approximately 50 free-floating spheroids were added to a 15 mL conical tube, and the supernatant was discarded; 1 mL of DMEM/F-12 with 15 mM HEPES was added to the settled spheroids and centrifuged at $300 \times g$ for 5

min at room temperature. The supernatant was discarded, and spheroids were resuspended in 50 µL of cold basement membrane matrix (Corning Matrigel Growth Factor Reduced Basement Membrane Matrix, Phenol Red-Free cat. no. 356231). Embedded spheroids were gently transferred into the center of one well of a Nunclon Delta surface treated 24-well tissue cultured plate and incubated at room temperature for 20 min. 500 µL of warmed OGM was added to the well and incubated at 37°C with 5% $CO_2$. Media were changed every 3–4 days and incubated at 37°C with 5% $CO_2$. After 7–10 days post-spheroid embedding in basement membrane matrix (dome formation), HIOs were passaged. It is important to note that the term "spheroids" is used specifically to describe the early-stage aggregates prior to full organoid maturation. This nomenclature aligns with the terminology provided by STEMCELL Technologies, which refers to these structures as spheroids during the initial phase of differentiation. At this stage, the cells have not yet developed the complex architecture or functional features characteristic of mature organoids. This term is intentionally used to reflect the developmental progression described in the manufacturer's protocol and distinguish between pre-organoid and organoid stages.

For HIO passages, the STEMCELL Technologies protocol was followed (39). One 15 mL conical tube per dome was coated with 2 mL of Anti-Adherence Rinsing Solution and rinsed with DPBS. Media were aspirated from the well with the dome; 1 mL of cold DMEM/F-12 was added and pipetted to mix the media with the dome. Cell suspension was transferred to a coated 15 mL conical tube, and a 1:3 ratio was used for passaging. The tube with cell suspension was then incubated on ice for about 5 min until organoid fragments settled to the bottom of the tube, and the supernatant was discarded. HIO pellet was resuspended in 2 mL of cold DMEM/F-12, centrifuged at $200 \times g$ for 5 min at room temperature, and the supernatant was discarded. HIO pellet was resuspended in 50 µL of cold basement membrane matrix as completed before and transferred to the center of one well of the previously mentioned 24-well plate. HIOs were incubated at room temperature for 20 min, and 500 µL of OGM was added to the well and incubated at 37°C with 5% $CO_2$. The medium was changed every 3–4 days and incubated at 37°C with 5% $CO_2$. Passages of HIOs were completed approximately every 7 days, depending on organoid density, size, and morphology (39).

## Apical-out HIE

This procedure was adapted from a previous study (29) with modifications (29). Apical-out HIEs were generated from established enteroids embedded in basement membrane matrix for at least 2–3 weeks from the initial plating from frozen stock. Media were removed, 600 µL of cold IntestiCult was added, and enteroids were collected using a 0.1% BSA-coated tip into a 0.1% BSA-coated 15 mL conical tube. The enteroids were gently triturated to remove basement membrane protein matrix and centrifuged at $300 \times g$ for 3 min at 4°C. The supernatant was removed without disrupting the cell pellet, and 1 mL of cold 5 mM EDTA in PBS was added; 9 mL of cold 5 mM EDTA in PBS was added to make a total volume of 10 mL. The tube was placed on a rotating platform in 4°C and incubated for 1 h with manual disruption every 20 min to solubilize the HIE. HIEs were centrifuged at $300 \times g$ for 3 min at 4°C, and the supernatant was removed without disrupting the cell pellet. The cells were resuspended in 5 mL of HCM. HIE were centrifuged again at $300 \times g$ for 3 min at 4°C, and the supernatant was removed. Cells were resuspended in 1 mL of HCM in 0.5 mL increments, and 11 mL of HCM was added to make a total of 12 mL cell suspension; 0.5 mL of cells suspended in HCM was added to each well of the ultra-low attachment 24-well plate and incubated at 37°C with 5% $CO_2$. On Day 2, the media were changed by collecting HIE using 0.1% BSA-coated tips into a 0.1% BSA-coated 15 mL conical tube. Cells were centrifuged at $300 \times g$ for 3 min at 4°C and resuspended in 1 mL of HCM by gently pipetting in 0.5 mL increments; 11 mL of HCM was added to the enteroids to make a total of 12 mL cell suspension, and 500 µL of HIE was transferred to each well of the ultra-low attachment 24-well plate. HIE began to visually reverse polarity by Day 2, and by Day 3, HIE was completely apical out. Day

3 is optimal for experimentation, with cell death occurring by Day 5. See Fig. 1A for the graphical illustration of the generation of apical-out enteroids.

## Apical-out HIO

The same protocol was replicated for apical-out HIO as apical-out HIE with the following modifications. Apical-out HIOs were generated from established organoids that had been passaged at least 2–3 times post-differentiation stages; 500 µL of cold OGM was added to each well, and HIOs were collected using tips coated with anti-adherence rinsing solution into a 15 mL conical tube coated with the same solution. HIOs were gently triturated to remove the basement membrane protein matrix and centrifuged at 300 × $g$ for 3 min at 4°C. The remaining of the HIO apical-out transformation was followed just as with the HIEs described previously. See Fig. 1A for graphical illustration of apical-out HIO generation.

## Microbial culture and exposure

GBS COH-1 (serotype III ST-17, BAA-1176) was used for all experiments and was grown overnight at 37°C in Tryptic Soy broth to the stationary phase. Both apical-out HIE and HIO were exposed to GBS at a concentration of $1.1 \times 10^7$ CFU/mL and $8.4 \times 10^7$ CFU/mL. For experiments using HIE monolayers, GBS was added at a concentration of $5 \times 10^7$ CFU/mL.

## Staining of apical-out HIE and HIO

On Day 3 from formation, apical-out HIEs/HIOs were exposed to GBS at concentrations of $1.1 \times 10^7$ CFU/mL and $8.4 \times 10^7$ CFU/mL and incubated for 2 h at 37°C with 5% $CO_2$. HIE/HIO were collected into 0.1% BSA-coated 1.5 mL microcentrifuge tubes and centrifuged at 300 × $g$ for 3 min at 4°C (each centrifuge step was completed using these criteria). The supernatant was removed, and HIE/HIO were resuspended in 500 µL of PBS for a total of three washes to retain only adhered and invaded GBS. Apical-out HIE/HIO were resuspended in 500 µL of 4% paraformaldehyde (PFA) and incubated for 30 min at room temperature. HIE/HIO was centrifuged, and the supernatant was removed. HIEs/HIOs were resuspended in blocking buffer (5% goat serum and 0.5% Triton X-100 in 1× PBS) for a total volume of 500 µL per microcentrifuge tube and incubated for 2 h at 4°C. Primary antibodies were prepared by diluting in blocking buffer (anti-Streptococcus Group B antibody, Abcam ab53584, 1:200 dilution; anti-Villin antibody, Abcam ab201989, 1:100 dilution). HIE/HIO was centrifuged, and the supernatant was removed. Each sample was resuspended in 500 µL of primary antibody solution and incubated overnight at 4°C. Secondary antibodies were prepared by diluting in blocking buffer (goat anti-rabbit IgG (H+L) Alexa Fluor 568, Fisher A-11036, 1:500 dilution; goat anti-mouse IgG Alexa Fluor 488, Abcam ab150113, 1:200 dilution). HIE/HIO were centrifuged, washed, and resuspended in 500 µL of secondary antibody and incubated overnight at 4°C. After washing, HIE/HIO were resuspended in DAPI (Fisher EN62248, 1:200 dilution) and incubated for 30 min at room temperature. Cells were centrifuged and resuspended in 100 µL of mounting solution (70% glycerol for HIEs and Prolong Gold Antifade Mountant without DAPI for HIOs). Slides were stored at 4°C prior to imaging. See Fig. 1B for the graphical illustration of the infection, staining, and imaging procedure of apical-out HIE/HIO. Figures 2H and 3H represent uninfected apical-out controls of HIE and HIO, respectively, for comparison purposes, and Fig. S1 confirms apical-out polarity reversal.

## Imaging

The Zeiss Laser Scanning Confocal Microscope 880 was used to capture z-stacked images collected at 0.5 µm intervals of HIEs and HIOs (100× and 63× oil objective, NA 1.4). Channels used were DAPI (nuclei, excitation at 405 nm, emission at ~461 nm), Alexa Fluor 488 (villin for the epithelial border or F-actin Phalloidin, excitation at 488 nm, emission at ~520 nm), and Alexa Fluor 568 (GBS, excitation at 578 nm, emission at

~603 nm). Surfaces were created in Imaris (v10.2, Oxford Instruments) using a fixed intensity threshold for each fluorescence channel. No manual smoothing or interpolation was applied beyond the default Gaussian filter (0.2 µm). The renderings are used solely for visualization of spatial relationships, not for quantitative analysis. For validation of cell type differentiation, HIO images were retained in their raw form, processed only with the default Gaussian filter (0.2 µm) to minimize background noise while preserving signal integrity.

## Gene expression

Apical-out HIE/HIO were exposed to GBS COH-1 at a concentration of $8.4 \times 10^7$ CFU/mL and incubated for 2 h at 37°C with 5% $CO_2$. Caco-2 cells were infected with GBS COH-1 at an MOI of 50 ($1 \times 10^8$ CFU/mL) and incubated for 2 h at 37°C with 5% $CO_2$. HIE, HIO, and Caco-2 RNA extractions were completed using the QIAGEN RNeasy Mini Kit and then purified with the Ambion DNase I (RNase-free) kit. cDNA was generated using the High-Capacity Reverse Transcription Kit (Applied Biosystems). Targeted qRT-PCR was performed using TaqMan Gene Expression Assays (Thermo Fisher Scientific). Glyceraldehyde 3-phosphate dehydrogenase (GAPDH) was used as the reference gene for both HIE and HIO samples, and β-actin was used for Caco-2 cells. Each cell type was analyzed under two conditions: untreated (GBS-untreated control) and GBS-exposed. HIEs included four biological replicates per condition with three technical replicates, while HIOs and Caco-2 cells included three biological replicates per condition with three technical replicates. The Ct values of the target genes were normalized to their respective housekeeping genes, and relative quantification was performed using the ΔΔ-CT method. The untreated control condition was used as the baseline reference, with its expression level normalized to 1.0 for $\log_2$ fold change calculations. All plotted data represent the $\log_2$ fold changes in expression of each target gene relative to this baseline. Validation of cell type presence in apical-out HIE was confirmed by qRT-PCR analysis (Fig. S2). To independently confirm the cell type composition of apical-out HIO, qRT-PCR analysis was performed of lineage-specific markers (Fig. S3).

## Generation of HIE monolayers

Generation of HIE monolayers was performed as previously described by our group (41) and from Michigan Protocols (32), which were adapted from previously established enteroid monolayer protocols (13, 30, 42–46). Transwell 24-well plates (Corning CLS3472) were coated with Collagen IV solution at a concentration of 33 µg/mL for 30 min at 4°C and then transferred to a 37°C incubator for 4 h. After incubation, collagen was removed.

Once HIE had been cultivated for 2–3 weeks, each well from 6-well plate was to be used for monolayer generation with the media changed to 1:1 Intesticult OGM:HCM the day before and morning of seeding. On the day of seeding, the media were aspirated, and 2 mL of cold 0.5 mM EDTA was added to each well and transferred to a 0.1% BSA-coated 15 mL conical tube. HIEs were triturated 5×, and the total volume was increased to a total of 10 mL EDTA. Enteroids were centrifuged at $200 \times g$ for 5 min at 4°C, and the supernatant was removed. Cells were resuspended with 1 mL (per well) of 0.05% Trypsin with 10 µM Y27632, triturated 5× with 1,000 µL 0.1% BSA-coated tip, and incubated for 3 min 45 s at 37°C with 5% $CO_2$. Trypsin was then inactivated with an equal amount of room temperature advanced DMEM with 10% FBS, 10 mM HEPES, and 10 µM Y27632 and triturated 50×. Cells were passed through a 40 µm cell strainer into a 50 mL 0.1% BSA-coated conical tube and centrifuged at $400 \times g$ for 5 min at 4°C. Supernatant was removed, and the cell pellet was resuspended in 1 mL of room temperature HCM. Cells were counted using the TC10 Automated Cell Counter (Bio-Rad) to produce 200,000 cell events at 100 µL per well. HIEs were seeded onto the top apical compartment of the Transwell plates with 3.0 µm pore size polyester membrane cell culture inserts (Corning CLS3472), and 600 µL of HCM media was added to the basolateral compartment (Day −1). The next day (Day 0), HCM was slowly aspirated, and 200 µL of 2D monolayer media (2DM) with 2.5 µM Y27632 was slowly added to each well. Media were then changed

every 2 days with 2DM (no Y27632). Transepithelial electrical resistance (TEER) was measured once daily using the Epithelial Volt/Ohm Meter (World Precision Instruments, EVOM2).

## Permeability, translocation, and cytotoxicity assay

Following HIE monolayer generation, once confluency was reached with a TEER value of ~200 $\Omega cm^2$, media were aspirated, and cells were washed with PBS three times. GBS was added at a concentration of $8 \times 10^7$ CFU and incubated for 2 h at 37°C with 5% $CO_2$. Positive controls for membrane permeability were 2-h incubation with 2 mM EGTA for disrupting the barrier without significant cell death, and Triton X-100 for disruption due to cell death. There were four conditions (untreated cells, GBS-exposed, 2 mM EGTA treatment, and Triton X-100 treatment) with four biological replicates for each condition. TEER measurements were recorded prior to and after exposure to GBS. Following 2-h incubations, cell viability was evaluated by collecting media from the apical compartment to measure the release of lactate dehydrogenase using the LDH-Glo Cytotoxicity assay kit (Promega Cat #J2380) and was followed according to the manufacturer's instructions. To determine the amount of translocated GBS, the media from the basolateral compartment were collected and enumerated by completing serial dilutions on GBS selection agar plates.

## Staining of the HIE monolayer

To visualize GBS interactions with HIE monolayers following 2-h exposure, the following protocol was adapted from a previous study (47) with modifications. Cells were washed three times with DPBS (without calcium and magnesium) and fixed with 4% paraformaldehyde for 15 min at room temperature. Cells were permeabilized with 0.2% Triton X-100 in DPBS for 10 min at room temperature, washed two times with DPBS containing 0.1% Tween 20 (PBST), and blocked for 2 h at room temperature in 10% goat serum, 0.1% BSA, and 0.01% Triton X-100 in DPBS. Cells were washed once with PBST and incubated overnight at 4°C in primary antibody for GBS (anti-Streptococcus Group B antibody, Abcam ab53584, 1:200 dilution). Cells were washed three times with PBST and incubated for 2 h at room temperature with secondary antibody (goat anti-rabbit IgG (H+L) Alexa Fluor 568, Fisher A-11036, 1:500 dilution) diluted in blocking buffer. Monolayers were washed three times with PBST and incubated with Alexa Fluor 488 Phalloidin (cat. A12379, Invitrogen) diluted 5:200 (1:40) in PBS for 30 min at room temperature. Washed once with PBST, incubated with DAPI for 30 min at room temperature, and washed twice with $dH_2O$. The membrane was removed using a scalpel and mounted on a microscope slide with VectaMount permanent mounting medium (cat. H-5000-60; Vector). Images were obtained as mentioned previously with apical-out HIE/HIO.

## Data analysis

All statistical analyses were performed using GraphPad Prism software. The unpaired $t$-test was used to compare gene expression data (expressed as $log_2$ fold change), and TEER measurements were compared using Kruskal-Wallis with Dunn's Correction. Cytotoxicity assay was evaluated using Kruskal-Wallis with Dunn's Correction.

## ACKNOWLEDGMENTS

HIE were supported by the NIH Award Number HL150300 (T.T.B.H.).

## AUTHOR AFFILIATIONS

[1]Pediatrics, Morsani College of Medicine, University of South Florida, Tampa, Florida, USA
[2]Molecular Medicine, Morsani College of Medicine, University of South Florida, Tampa, Florida, USA

## AUTHOR ORCIDs

Alexia N. Pearah  http://orcid.org/0009-0003-2629-828X
Nichol John-Lewis Edwards  http://orcid.org/0009-0002-1068-7582
Sophie E. Darch  http://orcid.org/0000-0003-2800-2844
Thao T. B. Ho  http://orcid.org/0000-0002-4580-9669
Tara M. Randis  http://orcid.org/0000-0003-0348-6766

## FUNDING

| Funder | Grant(s) | Author(s) |
| --- | --- | --- |
| National Institutes of Health | HL150300 | Thao T. B. Ho |

## AUTHOR CONTRIBUTIONS

Alexia N. Pearah, Conceptualization, Data curation, Formal analysis, Investigation, Methodology, Validation, Visualization, Writing – original draft, Writing – review and editing | Nichol John-Lewis Edwards, Data curation, Methodology, Writing – review and editing | Rico R. Carter, Data curation, Methodology | Leigh-Anne M. Worthington, Data curation, Methodology | Madison R. Huszar, Data curation, Methodology, Writing – review and editing | Kristen Domínguez, Data curation, Methodology, Writing – original draft, Writing – review and editing | Hannah L. Wapshott-Stehli, Methodology, Writing – review and editing | April K. Lindon, Investigation, Methodology, Writing – review and editing | Sophie E. Darch, Conceptualization, Data curation, Formal analysis, Investigation, Methodology, Resources, Software, Supervision, Validation, Visualization, Writing – original draft, Writing – review and editing | Thao T. B. Ho, Conceptualization, Data curation, Formal analysis, Funding acquisition, Investigation, Methodology, Resources, Supervision, Validation, Visualization, Writing – original draft, Writing – review and editing | Tara M. Randis, Conceptualization, Data curation, Formal analysis, Funding acquisition, Investigation, Methodology, Project administration, Resources, Software, Supervision, Validation, Visualization, Writing – original draft, Writing – review and editing

## ADDITIONAL FILES

The following material is available online.

### Supplemental Material

**Fig. S1 (Spectrum02316-25-S0001.tif).** Polarity reversal controls.
**Fig. S2 (Spectrum02316-25-S0002.tif).** HIE cell type validation.
**Fig. S3 (Spectrum02316-25-S0003.tif).** HIO cell type validation.
**Supplemental material (Spectrum02316-25-S0004.docx).** Supplemental figure legends.

### Open Peer Review

**PEER REVIEW HISTORY (review-history.pdf).** An accounting of the reviewer comments and feedback.

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
