## [Reviewer comments · Microbiology Spectrum]

Microbiology Spectrum

Bridging the Gap: Organotypic Models to Study Late-onset Group B Streptococcus Infection

Alexia Pearah, Nichol John-Lewis Edwards, Rico Carter, Leigh-Anne Worthington, Madison Huszar, Kristen Dominguez, Hannah Wapshott-Stehli, April Lindon, Sophie Darch, Thao Ho, and Tara Randis

Corresponding Author(s): Alexia Pearah, University of South Florida Morsani College of Medicine

Review Timeline:

Submission Date:	July 30, 2025
Editorial Decision:	September 2, 2025
Revision Received:	November 3, 2025
Editorial Decision:	December 2, 2025
Revision Received:	February 16, 2026
Accepted:	March 25, 2026

Editor: Nagendran Tharmalingam

Reviewer(s): Disclosure of reviewer identity is with reference to reviewer comments included in decision letter(s). The following individuals involved in review of your submission have agreed to reveal their identity: Raghavendra Nagampalli (Reviewer #2)

Transaction Report:

DOI: <https://doi.org/10.1128/spectrum.02316-25>

Re: Spectrum02316-25 (Bridging the Gap: Organotypic Models to Study Late-onset Group B Streptococcus Infection)

Dear Dr. Alexia N Pearah:

Thank you for the privilege of reviewing your work. Below you will find my comments, instructions from the Spectrum editorial office, and the reviewer comments.

Revision Guidelines

Sincerely,
Nagendran Tharmalingam
Editor
Microbiology Spectrum

Reviewer #1 (Comments for the Author):

Brief description of the paper:

The authors describe the use of apical-out intestinal organoids either derived from human fetal tissue or from iPSC line to model Group B Streptococcus infection.

Major Comments:

- 1] While the use of fetal tissue-derived organoids or organoids derived from iPSCs is interesting, I am lacking important information as well as controls. The study needs to include controls such as GBS-uninfected organoids etc. Perhaps they were performed but are not shown here, and the manuscript would benefit from a direct comparison.
- 2] It is not clear what the 3D visualization based on the confocal image adds to the data. Why is this type of visualization used? How was it rendered? The experimental information lacks sufficient details and the low quality of the original microscopy images leads to strange artefacts including merging nuclei of neighboring cells into one (e.g. Figure 2). Since microscopy data are a key portion of the paper, this is vital information.
- 3] In Figure S1, we lack a clear comparison of staining by different markers for the same organoid. Without a brightfield image or at least a nuclear staining image, it is impossible to assess the images, especially for panel D but also B. Ideally, we would look at a representative image of one organoid stained by multiple markers and then at a merged image.
- 4] GBS invasion into organoids and cells - since authors want to make conclusions about the invasion of GBS, we need to see clearer data. For the intraorganoid invasion, we would need to see a confocal image going through an organoid cavity. 1D might look like the GBS is inside, but it is hard to make a conclusion since the section might be going through a surface layer. There is a lot of noise from villin antibody outside of the organoid itself, that the visualization in 1D removes, thereby "improving" the actually observed staining. Video in 1F only includes a part of an organoid, so we are not sure if the organoid was otherwise intact. For the intracellular localization - the visualization in 2D,F does not necessarily show intracellular localization and as described in my previous question - it is a 3D rendered visualization that appears to have artifacts.
- 5] I think the publication would benefit from a brief explanation as to why did the expression data include these particular genes? How do these changes compare to other literature on transcriptomic changes after GBS infection e.g.
<https://doi.org/10.1128/iai.00035-23> ?
- 6] The organoids shown in microscopy don't appear to have a cavity. Is this representative of all organoids observed in the study?

Minor Comments:

- 1] Labeling is not consistent throughout the paper. Organoids are labeled as HIE/HIO, but then Figure 3 switches to S.I. Enteroids and Organoids. Caco-2 cells are labeled as CaCO₂ (chemical compound?) in multiple spots including Figure 3, but also in the text of the paper. GBS is labeled as COH1 in Figure 4. I understand that that is the particular strain used, but for the sake of clarity, I would recommend using GBS as is used in the Figure 4 legend as well in the other cases where GBS is mentioned (e.g. other figures).
- 2] Scale bars - in Figure 1, the scale bars do not have a legible description and it is not clear what they corresponds to.
- 3] Is 1A a zoomed in image?
- 4] Image 2C seems to be compressed from the sides.
- 5] Labeling in Figure 3 is incorrect. The legend description does not correspond to the plots shown in A, B and C. Perhaps the plots were shuffled around and the legend should be adjusted accordingly.
- 6] Row 176 - *in vivo* should be italicized
- 7] Human colonoid medium - please describe the actual medium. First two references Dame et al. and Miyoshi and Stappenbeck do not describe "human colonoid medium". It is not clear what medium was used. I was not able to find a Michigan Protocols human colonoid medium.
- 8] The sentence on line 240 duplicates previous sentence, was the basement matrix Matrigel? Please, merge the two sentences to include all information for the sake of clarity. The basement matrix mentioned on line 246 is also Matrigel?
- 9] Line 281 - what factors were added to the medium? Only Y-27632 or others as well? What was the source of Y-27632?
- 10] Polarity reversal protocol, line 310 - how many days after passaging was the polarity reversal performed? Could the authors include the images on polarity reversal comparison between day 0 and 3 to essentially compare unreversed vs reversed organoids? This would be of value since the polarity reversal appears to be slower compared to adult enteroid shown in Co et al.
- 11] The schemes in Figure 5 are useful, but could perhaps be moved in the manuscript to Figure 1 to explain the workflow in the manuscript as sort of a graphical summary?
- 12] I would avoid the use of the term spheroids, since those are different from organoids.
- 13] Line 367 - Dapi should be DAPI.
- 14] The expression data should include a GBS-untreated control and a detailed explanation about normalizations done in the methods section. Currently, it is not clear how the data was processed before being plotted.

Reviewer #2 (Comments for the Author):

In this work, authors have used human tissue-derived enteroids (HIEs) and pluripotent stem cell-derived organoids (HIOs) to model Group B Streptococcus (GBS). In their findings they showed that GBS alters the transcriptome and increases barrier permeability in these models, which accurately reflect neonatal physiology than traditional immortal cell lines. GBS was observed to attach to and, in some cases, translocate across the intestinal barrier. These results highlight HIEs and HIOs as valuable platforms for studying late-onset GBS infection and for testing new preventative strategies. This is a useful contribution and can be accepted for publication upon addressing the following

1. Did authors test the co-cultures of HIEs/HIOs with other innate immune cells (e.g., macrophages, neutrophils)?
2. What is the cell death mechanism (apoptosis, necrosis, or other forms of cell death) in intestinal epithelium upon exposure to

GBS

3. What are the major gene networks that are upregulated or downregulated in response to GBS in the organoid models, and how do these differ from those in immortalized cell lines?

Comments and Suggestions for the Author:

Brief description of the paper:

The authors describe the use of apical-out intestinal organoids either derived from human fetal tissue or from iPSC line to model Group B Streptococcus infection.

Major Comments:

1] While the use of fetal tissue-derived organoids or organoids derived from iPSCs is interesting, I am lacking important information as well as controls. The study needs to include controls such as GBS-uninfected organoids etc. Perhaps they were performed but are not shown here, and the manuscript would benefit from a direct comparison.

2] It is not clear what the 3D visualization based on the confocal image adds to the data. Why is this type of visualization used? How was it rendered? The experimental information lacks sufficient details and the low quality of the original microscopy images leads to strange artefacts including merging nuclei of neighboring cells into one (e.g. Figure 2). Since microscopy data are a key portion of the paper, this is vital information.

3] In Figure S1, we lack a clear comparison of staining by different markers for the same organoid. Without a brightfield image or at least a nuclear staining image, it is impossible to assess the images, especially for panel D but also B. Ideally, we would look at a representative image of one organoid stained by multiple markers and then at a merged image.

4] GBS invasion into organoids and cells - since authors want to make conclusions about the invasion of GBS, we need to see clearer data. For the intraorganoid invasion, we would need to see a confocal image going through an organoid cavity. 1D might look like the GBS is inside, but it is hard to make a conclusion since the section might be going through a surface layer. There is a lot of noise from villin antibody outside of the organoid itself, that the visualization in 1D removes, thereby “improving” the actually observed staining. Video in 1F only includes a part of an organoid, so we are not sure if the organoid was otherwise intact. For the intracellular localization - the visualization in 2D,F does not necessarily show intracellular localization and as described in my previous question – it is a 3D rendered visualization that appears to have artifacts.

5] I think the publication would benefit from a brief explanation as to why did the expression data include these particular genes? How do these changes compare to other literature on transcriptomic changes after GBS infection e.g.

<https://doi.org/10.1128/iai.00035-23> ?

6] The organoids shown in microscopy don't appear to have a cavity. Is this

representative of all organoids observed in the study?

Minor Comments:

1] Labeling is not consistent throughout the paper.

Organoids are labeled as HIE/HIO, but then Figure 3 switches to S.I. Enteroids and Organoids.

Caco-2 cells are labeled as CaCO₂ (chemical compound?) in multiple spots including Figure 3, but also in the text of the paper.

GBS is labeled as COH1 in Figure 4. I understand that that is the particular strain used, but for the sake of clarity, I would recommend using GBS as is used in the Figure 4 legend as well in the other cases where GBS is mentioned (e.g. other figures).

2] Scale bars – in Figure 1, the scale bars do not have a legible description and it is not clear what they corresponds to.

3] Is 1A a zoomed in image?

4] Image 2C seems to be compressed from the sides.

5] Labeling in Figure 3 is incorrect. The legend description does not correspond to the plots shown in A, B and C. Perhaps the plots were shuffled around and the legend should be adjusted accordingly.

6] Row 176 – *in vivo* should be italicized

7] Human colonoid medium – please describe the actual medium. First two references Dame et al. and Miyoshi and Stappenbeck do not describe “human colonoid medium”. It is not clear what medium was used. I was not able to find a Michigan Protocols human colonoid medium.

8] The sentence on line 240 duplicates previous sentence, was the basement matrix Matrigel? Please, merge the two sentences to include all information for the sake of clarity. The basement matrix mentioned on line 246 is also Matrigel?

9] Line 281 – what factors were added to the medium? Only Y-27632 or others as well? What was the source of Y-27632?

10] Polarity reversal protocol, line 310 – how many days after passaging was the polarity reversal performed? Could the authors include the images on polarity reversal comparison between day 0 and 3 to essentially compare unreversed vs reversed organoids? This would be of value since the polarity reversal appears to be slower

compared to adult enteroid shown in Co et al.

11] The schemes in Figure 5 are useful, but could perhaps be moved in the manuscript to Figure 1 to explain the workflow in the manuscript as sort of a graphical summary?

12] I would avoid the use of the term spheroids, since those are different from organoids.

13] Line 367 – Dapi should be DAPI.

14] The expression data should include a GBS-untreated control and a detailed explanation about normalizations done in the methods section. Currently, it is not clear how the data was processed before being plotted.

Do NOT indicate whether the paper should be accepted or rejected.

Confidential remarks for the Editors:

In its current state, in my opinion the study lacks some important information and controls making it hard to assess the experimental and methodological rigor. If the authors revise the data and methods section, I believe the paper might be suitable for publication.

I would recommend not using rendering for confocal microscopy images in the manner used here, as I am not sure what is the added benefit and worry about the distortion of the original results, but maybe the authors have a good justification of the method used.

In this work, authors have used human tissue-derived enteroids (HIEs) and pluripotent stem cell-derived organoids (HIOs) to model Group B Streptococcus (GBS). In their findings they showed that GBS alters the transcriptome and increases barrier permeability in these models, which accurately reflect neonatal physiology than traditional immortal cell lines. GBS was observed to attach to and, in some cases, translocate across the intestinal barrier. These results highlight HIEs and HIOs as valuable platforms for studying late-onset GBS infection and for testing new preventative strategies. This is a useful contribution and can be accepted for publication upon addressing the following

1. Did authors test the co-cultures of HIEs/HIOs with other innate immune cells (e.g., macrophages, neutrophils)?
2. What is the cell death mechanism (apoptosis, necrosis, or other forms of cell death) in intestinal epithelium upon exposure to GBS
3. What are the major gene networks that are upregulated or downregulated in response to GBS in the organoid models, and how do these differ from those in immortalized cell lines?

Reviewer #1:

Major Comments:

1] While the use of fetal tissue-derived organoids or organoids derived from iPSCs is interesting, I am lacking important information as well as controls. The study needs to include controls such as GBS-uninfected organoids etc. Perhaps they were performed but are not shown here, and the manuscript would benefit from a direct comparison.

We appreciate your interest in our organoid models. We have added representative images of uninfected HIE in Figure 2G and HIO in Figure 3G. Uninfected controls are included for all qPCR experiments as well.

2] It is not clear what the 3D visualization based on the confocal image adds to the data. Why is this type of visualization used? How was it rendered? The experimental information lacks sufficient details and the low quality of the original microscopy images leads to strange artefacts including merging nuclei of neighboring cells into one (e.g. Figure 2). Since microscopy data are a key portion of the paper, this is vital information.

We thank the reviewer for this thoughtful comment. We have clarified both the rationale and the technical details of the 3D visualization. The purpose of the Imaris renderings is to illustrate the spatial organization of cells and extracellular components within the organoid, which cannot be fully appreciated in 2D images or single optical planes. The 3D reconstruction provides volumetric context—highlighting the arrangement of peripheral cell layers surrounding a hollow lumen characteristic of mature organoids.

The 3D renderings were generated from confocal z-stacks collected at 0.5 μm intervals using a Zeiss LSM880 microscope (63 \times oil objective, NA 1.4). Surfaces were created in Imaris (v10.2, Oxford Instruments) using a fixed intensity threshold for each fluorescence channel (blue = nuclei, DAPI; green = villin for the epithelial border, anti-villin IgG Alexa Fluor™ 488; magenta = GBS, anti-GBS IgG Alexa Fluor™ 568). No manual smoothing or interpolation was applied beyond the default Gaussian filter (0.2 μm). The renderings are used solely for visualization of spatial relationships, not for quantitative analysis.

Regarding the appearance of merged nuclei, this effect results from the projection plane of the static image rather than from segmentation or rendering artefacts. Because the figure represents a single 3D orientation, overlapping nuclei along the z-axis can appear continuous in projection. We have added a supplementary rotational video (Figure 2D QR code in lower right quadrant) that demonstrates the full 3D structure of HIE, clearly showing discrete nuclei surrounding a central hollow core. We have also revised the figure legend and methods section to explain the

rendering parameters and imaging geometry in greater detail. Please note that previous figure 2 is now figure 3 in the revised manuscript and legend revised as noted below.

Figure 3. GBS adherence to apical-out epithelial border of human intestinal organoids.

Representative images using laser confocal scanning microscopy at (A) 63x magnification, (B) 1.5x zoomed in at 63x magnification, thus image is at 94.5x magnification (C) 100x magnification. Three-dimensional rendering of organoid architecture from confocal z-stacks images acquired using a Zeiss LSM880 microscope equipped with a 63× oil objective). Optical sections were collected at 0.5 μm intervals and rendered in Imaris (v10.2, Oxford Instruments) using the *Surface* tool with fixed intensity thresholds applied to each fluorescence channel [SD1] (D) staining of GBS [magenta = anti-GBS IgG Alexa Fluor 568], (E) nuclei [Blue = DAPI], (F) Composite image. (G) Uninfected HIO control image included for reference alongside GBS-infected HIO. Scale bars: Panel A, 20 μm; Panel B, 10 μm; Panel C, 5 μm; Panel D-F, 25 μm; Panel G, 10 μm.

3] In Figure S1, we lack a clear comparison of staining by different markers for the same organoid. Without a brightfield image or at least a nuclear staining image, it is impossible to assess the images, especially for panel D but also B. Ideally, we would look at a representative image of one organoid stained by multiple markers and then at a merged image.

We thank the reviewer for this helpful comment. We wish to clarify that the images in Figure S1 were direct visualizations of the raw confocal volumes. After reviewing the data, we determined that rendering these images would not improve clarity and could instead introduce artifacts or smoothing effects that obscure true staining patterns. Therefore, the images were retained in their raw form, processed only with the default Gaussian filter (0.2 μm) to minimize background noise while preserving signal integrity.

The iPSC-derived organoids shown in Figure S1 are structurally fragile, which limits the number of markers that can be reliably stained in a single section without compromising antigenicity. To represent key aspects of organoid organization, we selected the most robust stains across serial sections and included nuclear (DAPI) channels where available to aid orientation and interpretation.

To independently confirm cell-type composition, we have also performed PCR analysis of lineage-specific markers corresponding to the antigens visualized by immunostaining (Figure S1 E). We have updated the figure legend and methods section to clarify the imaging approach and the use of Gaussian filtering, and to emphasize that these panels represent distinct sections from iPSC-derived organoids.

4] GBS invasion into organoids and cells - since authors want to make conclusions about the

invasion of GBS, we need to see clearer data. For the intraorganoid invasion, we would need to see a confocal image going through an organoid cavity. 1D might look like the GBS is inside, but it is hard to make a conclusion since the section might be going through a surface layer. There is a lot of noise from villin antibody outside of the organoid itself, that the visualization in 1D removes, thereby "improving" the actually observed staining. Video in 1F only includes a part of an organoid, so we are not sure if the organoid was otherwise intact. For the intracellular localization - the visualization in 2D,F does not necessarily show intracellular localization and as described in my previous question - it is a 3D rendered visualization that appears to have artifacts.

We thank the reviewer for this detailed feedback and the opportunity to clarify the imaging approach. The images shown in the previously labeled Figure 1D–F (now Figure 2B–D) are indeed 3D renderings generated from complete confocal z-stacks using Imaris (v10.2, Oxford Instruments). The purpose of this rendering is to illustrate the spatial localization of *Group B Streptococcus* (GBS) within the three-dimensional architecture of the organoid — information that cannot be fully appreciated from single optical sections.

To minimize artefacts, surfaces were generated using the *Surface* tool with fixed, channel-specific intensity thresholds and the default Gaussian filter (0.2 μm) to reduce background noise. No manual interpolation or smoothing was applied. The underlying confocal stacks span the entire organoid, including the central lumen, and confirm that GBS signal extends into internal regions rather than being restricted to the surface. The static image in Figure 1D (now Figure 2B) represents one orientation within the full 3D volume, which can give the impression of surface localization when viewed in isolation.

We have now included an additional supplementary video showing the full rotation and z-projection of the same organoid (Figure 2D lower right quadrant), demonstrating both the intact structure and bacterial presence within the internal cavity and epithelial cell layer. Apparent background signal from villin outside the organoid arises from detached cell fragments and luminal debris typical of infected samples, not from rendering artefacts.

We have revised the figure legend and Methods section to explicitly describe the rendering parameters, the inclusion of the full z-stack volume, and the use of Gaussian filtering. Together, these clarifications provide transparency regarding data processing and demonstrate that the 3D visualization faithfully represents bacterial localization within intact organoids.

5] I think the publication would benefit from a brief explanation as to why did the expression data include these particular genes? How do these changes compare to other literature on transcriptomic changes after GBS infection e.g. <https://doi.org/10.1128/iai.00035-23> ?

We thank the reviewer for the opportunity to expand on this topic. Previous studies by our group using both a mouse model of GBS intestinal colonization as well as Caco-2 cell lines revealed

GBS-induced alterations in IEC gene expression — specifically in those related to barrier function. We conducted a transcriptomic analysis of the mouse model that revealed changes in the KEGG pathway responsible for epithelial barrier structure and function and validated those findings with Caco-2 cells through targeted qRT-PCR. This study prioritized the selected genes based on importance to epithelial barrier function upon GBS exposure across the models of Caco-2 cells, HIEs, and HIOs. We have updated the discussion section to include this explanation.

6] The organoids shown in microscopy don't appear to have a cavity. Is this representative of all organoids observed in the study?

We thank the reviewer for this observation. The images presented in the main figures are apical-out HIE/HIO that do not have a central cavity/lumen after completing the polarity reversal process. This morphology is representative of all apical out HIE/HIO in this study, however, during standard growth conditions of HIE/HIO before polarity reversal, the apical surface is faced inward creating the central cavity/lumen. As described in the manuscript, in this configuration access to the apical surface is limited, making apical out configured HIE/HIO ideal for visualizing direct host-pathogen interactions. The process of generating these apical out HIE/HIO, collapses the central lumen, causing the HIE/HIO to become spheroid-like in structure with the apical surface exposed externally. To clarify this process, the following was revised in the introduction: “The luminal surface of enterocytes, which is enclosed within the interior of HIE and HIO spheroids, is difficult to access within the central cavity. To overcome this challenge, polarity reversal of HIEs/HIOs is generated to produce apical-out HIEs and HIOs, thus collapsing the central cavity and allowing for the luminal apical surface of IECs to be in direct contact with reagents and/or microbes added to culture media [6].”

Minor Comments:

1] Labeling is not consistent throughout the paper.

Organoids are labeled as HIE/HIO, but then Figure 3 switches to S.I. Enteroids and Organoids. Caco-2 cells are labeled as CaCO2 (chemical compound?) in multiple spots including Figure 3, but also in the text of the paper.

GBS is labeled as COH1 in Figure 4. I understand that that is the particular strain used, but for the sake of clarity, I would recommend using GBS as is used in the Figure 4 legend as well in the other cases where GBS is mentioned (e.g. other figures).

Resolved in figures and figure legend.

2] Scale bars - in Figure 1, the scale bars do not have a legible description, and it is not clear what they correspond to.

We thank the reviewer for this observation. Note that original Figure 1 is now Figure 2. Scale bars indicate the magnification and in Figure 2, represent 3 μm for Panel A, 10 μm for Panels B-F, and Panel G, 15 μm . Description has been added to the figure legend for both figures 2 and 3.

3] Is 1A a zoomed in image?

Figure 1A (now Figure 2A) is a zoomed in image as indicated by the scale bars. Figure 2A has a scale bar of 3 μm and the other panels in figure 2 have a scale bar of 10 μm . Figure 2 description has been updated as seen below:

Figure 2. Laser confocal scanning microscopy imaging of GBS-exposed HIE. (A) Increased magnification of GBS adhering to epithelial cell surface. **(B)** GBS is visualized within the core of apical-out enteroid. **(C)** 3D rendering of image D. **(D)** Two QR codes link to 3D Vimeo videos: The upper-left video displays nuclei and GBS rendered with raw filter of Villin, allowing enhanced visualization of internalized GBS while the lower-right video includes all components rendered—nuclei, GBS, and Villin. **(E-F)** GBS adhesion and invasion of HIE enteroid using 3D rendering of confocal images. **(G)** Representative image of HIE control (no GBS exposure) shown for contextual comparison. [blue = nuclei, DAPI; green = villin, anti-villin IgG Alexa Fluor™ 488; magenta = GBS, anti-GBS IgG Alexa Fluor™ 568] Scale bars: Panel A, 3 μm ; Panel B-F, 10 μm ; Panel G, 15 μm .

4] Image 2C seems to be compressed from the sides.

We thank the reviewer for bringing this to our attention. Note original Figure 2C is now Figure 3C. Image has been corrected.

5] Labeling in Figure 3 is incorrect. The legend description does not correspond to the plots shown in A, B and C. Perhaps the plots were shuffled around and the legend should be adjusted accordingly.

We thank the reviewer for finding this error. Note original Figure 3 is now Figure 4. Figures and figure legends have been updated.

6] Row 176 - in vivo should be italicized

Resolved in manuscript.

7] Human colonoid medium - please describe the actual medium. First two references Dame et

al. and Miyoshi and Stappenbeck do not describe "human colonoid medium". It is not clear what medium was used. I was not able to find a Michigan Protocols human colonoid medium.

We thank the reviewer for this comment and the opportunity to clarify. The full composition of the human colonoid medium (HCM) is described in detail in Reference 32, which includes a direct link to the Michigan Protocols. Within that resource, item #5 (“Complete gastrointestinal growth media [HCM...]*”) links to a Google document authored by Dame et al. that provides a comprehensive breakdown of the media components and preparation steps.

As noted in the manuscript, the Michigan protocol was adapted and further optimized using additional published work (References 34, 36-38). For example, Reference 37 (Miyoshi and Stappenbeck) describes the generation and formulation of LWRN-conditioned media, which comprises 65% of the HCM formulation. While we follow the Michigan protocol for HCM media, we also wish to acknowledge the foundational work from which that protocol was derived, as these sources informed the optimization of HCM for organotypic intestinal models.

8] The sentence on line 240 duplicates previous sentence, was the basement matrix Matrigel? Please, merge the two sentences to include all information for the sake of clarity. The basement matrix mentioned on line 246 is also Matrigel?

We thank the reviewer for this observation. All basement membrane matrix used for HIO was Matrigel. Sentence has been updated in the manuscript and now reads “16 μ L of basement membrane (Matrigel hESC –Qualified Matrix (Corning) diluted to a concentration of 2%) was combined with 1 mL of cold DMEM/F12 and 1 mL of the mixture was used to coat each well of the 6 well plate.

9] Line 281 - what factors were added to the medium? Only Y-27632 or others as well? What was the source of Y-27632?

We thank the reviewer for allowing us the opportunity to clarify line 281. The medium used for Day 10, stage 3 organoid formation was Intestinal Organoid Growth Medium by combining STEMdiff™ Intestinal Organoid Basal Medium and STEMdiff™ Intestinal Organoid Supplement as described in the Stem Cell Technologies protocol. No other additives were included. Methods have been updated to clarify.

10] Polarity reversal protocol, line 310 - how many days after passaging was the polarity reversal performed? Could the authors include the images on polarity reversal comparison between day 0 and 3 to essentially compare unreversed vs reversed organoids? This would be of value since the polarity reversal appears to be slower compared to adult enteroid shown in Co et al.

We thank the reviewer for the valuable feedback. Polarity reversal was initiated 5 to 7 days after passaging, once the HIE had re-established structural integrity. While we appreciate the suggestion to include comparative images from days 0 to 3, we respectfully note that the HIE are highly fragile during the early stages of reversal and do not tolerate fixation and staining well at that time. As such, imaging at these time points was not feasible. Polarity reversal was confirmed through staining for villin localized to the apical surface, as shown in the representative images provided. These images reflect the successful establishment of apical-out polarity and serve as the basis for validating the reversal.

11] The schemes in Figure 5 are useful, but could perhaps be moved in the manuscript to Figure 1 to explain the workflow in the manuscript as sort of a graphical summary?

We appreciate the suggestion to reposition the schematic in Figure 5 as a graphical summary earlier in the manuscript and has now been moved to **Figure 1**. The following sentence was added to the Results section: A schematic in **Figure 1A** illustrates the generation of HIE/HIO and **Figure 1B** depicts the subsequent immunostaining procedure.

12] I would avoid the use of the term spheroids, since those are different from organoids.

We thank the reviewer for this comment and the opportunity to clarify why this term is used in this context. In this manuscript, the term “spheroids” is used specifically to describe the early-stage aggregates prior to full organoid maturation. This nomenclature aligns with the terminology provided by Stemcell Technologies, which refers to these structures as spheroids during the initial phase of differentiation. At this stage, the cells have not yet developed the complex architecture or functional features characteristic of mature organoids. We have used this term intentionally to reflect the developmental progression described in the manufacturer’s protocol and to distinguish between pre-organoid and organoid stages. The Methods section has been updated to convey this description.

13] Line 367 - Dapi should be DAPI.

Resolved in manuscript.

14] The expression data should include a GBS-untreated control and a detailed explanation about normalizations done in the methods section. Currently, it is not clear how the data was processed before being plotted.

We thank the reviewer for the opportunity to clarify the GBS-untreated control. Targeted qRT-PCR was performed using TaqMan™ Gene Expression Assays (Thermo Fisher Scientific). Glyceraldehyde 3-phosphate dehydrogenase (GAPDH) was used as the reference gene for both

HIE and HIO samples, and β -actin was used for Caco-2 cells. Each cell type was analyzed under two conditions: untreated (GBS-untreated control) and GBS-exposed. HIEs included four biological replicates per condition with three technical replicates, while HIOs and Caco-2 cells included three biological replicates per condition with three technical replicates. The Ct values of the target genes were normalized to their respective housekeeping genes, and relative quantification was performed using the $\Delta\Delta$ -CT method. The untreated control condition was used as the baseline reference, with its expression level normalized to 1.0 for \log_2 fold change calculations. All plotted data represent the \log_2 fold changes in expression of each target gene relative to this baseline. The Methods section and figure legend have been revised to reflect this explanation.

Reviewer #2:

1] Did authors test the co-cultures of HIEs/HIOs with other innate immune cells (e.g., macrophages, neutrophils)?

We thank the reviewer for their shared interest in co-culture models with HIE/HIO. We have not yet performed co-culture experiments involving HIEs/HIOs with innate immune cells such as macrophages or neutrophils. However, we are actively developing a model within our laboratory to investigate the interactions between HIEs/HIOs and macrophages in the context of Group B Streptococcus (GBS) exposure, with the goal of elucidating epithelial-immune dynamics during GBS adhesion, invasion, translocation and permeability.

2] What is the cell death mechanism (apoptosis, necrosis, or other forms of cell death) in intestinal epithelium upon exposure to GBS?

In this work we did not characterize cell death mechanisms triggered by GBS challenge in this manuscript. The literature would suggest that GBS-induced epithelial death is likely mostly programmed cell death (apoptosis, pyroptosis, or necroptosis) at physiologic infection levels but may also be mediated by direct lysis when exposed to high concentrations of the GBS pore forming toxin β -Hemolysin/Cytolysin. We measured cell viability in our HIE monolayers and noted it was greater than 80% cell viability following GBS exposure for 2 hours.

3] What are the major gene networks that are upregulated or downregulated in response to GBS in the organoid models, and how do these differ from those in immortalized cell lines?

We appreciate the reviewer's interest in the transcriptional responses to GBS in intestinal organoid models. However, this manuscript is focused on methodological development and does not include transcriptomic analyses. RNA-seq was not performed as part of this study, and characterization of gene networks upregulated or downregulated in response to GBS exposure is beyond the scope of the current work. Observed differences between selected genes of interest

suggest differential response that is dependent on model type (transformed, HIE, HIO). Future studies incorporating RNA-seq will be essential to elucidate these differential gene expression profiles and further define the molecular mechanisms underlying host-pathogen interactions in physiologically relevant models.

Re: Spectrum02316-25R1 (Bridging the Gap: Organotypic Models to Study Late-onset Group B Streptococcus Infection)

Dear Dr. Alexia N Pearah:

I received a comment, "I would recommend asking the authors to show high quality confocal microscopy data to establish whether the HIE can undergo polarity reversal and mimic the preterm intestinal tissue (actin, nuclei stained at minimum to confirm apical-out polarity). Authors should also amend their statements about transcriptomics and add important controls e.g. positive control for induced permeability in the TEER experiment for organoid-derived monolayers"

After careful consideration, instead of rejection, we are inviting you to modify the manuscript according to the reviewer's request.

If you think the reviewer's request is not doable, show me the reason, and please upload a response to the reviewer's comments. I will see what I can do further.

Revision Guidelines

Sincerely,
Nagendran Tharmalingam
Editor
Microbiology Spectrum

Reviewer #1 (Comments for the Author):

Major comments:

Main focus of this study was whether apical-out HIEs or HIOs could serve as a representative model for GBS infection of pre-term infants. However, the data shown is not sufficient to support the conclusions of the study. The only data shown to describe the polarity reversal and impact on different cell types and tissue structures lacks important information e.g. no brightfield image or Hoechst staining for the markers in Figure S1.

Furthermore, authors claim they observed changes in the transcriptome upon GBS infection but only couple of genes were studied. The expression levels were normalized to different housekeeping genes for organoids and for a cell line making it hard to draw direct comparisons. The appropriate control for the organoids described here would be expression data for actual tissue samples, not from a cell line, to ensure that organoids model the situation well.

Number of papers appear to be cited in a confusing way. E.g. on line 164 the citation 20 does not describe comparison of expression with Caco-2 cells or demonstrate it as a suitable control for in vivo expression data. It also does not suggest that the genes CLDN2, NOS2, OLFM4 are the ones with key differences between GBS infected and uninfected mice. The authors don't show transcriptomic analysis they claim to have done for HIE, HIO and Caco-2 and therefore it is not clear what they base their selection of the genes shown on. Line 113 - the reference 5 does not show cell marker staining for apical-out organoids since those are not used in the cited publication at all. Reference 5 describes using fetal-tissue derived organoids for microinjections - without polarity reversal. For HIO, images in Figure S1 are missing DAPI staining and/or brightfield image and the reader cannot evaluate whether the polarity reversal occurred the way authors claim it did. On line 169 - references 14 and 21 don't describe differences of organoids with Caco-2 and T-84 cell lines.

Figure S1 in general: The figure is missing nuclear staining or a brightfield picture to accompany the marker staining. This does not require any additional sample preparation that would impact the organoids, yet, it is not included. Therefore, the reader cannot assess the staining of the markers at all. This is especially true for Figure S1C, but for applies to all of them. Subfigure E then compares expression levels of some markers between a cell line Caco-2 and iPSC derived organoids, however, I don't think the expectation would be that they are the same. No data are shown for the HIE. The conclusion that there are "changes to the transcriptome" is not justified in my opinion. If anything, 6 specific genes are shown, 4 of them with a significant change in expression in HIO over Caco-2, but that is not representative of a transcriptome-wide shift and it is unclear if one or the other models the situation in preterm neonatal intestine.

What is the reasoning to use HIOs for modelling immature preterm neonatal intestine and its interaction with GBS?

Figure 5A - TEER measurement - this experiment lacks positive control for induced monolayer permeability.

Additional minor comments:

Figure 1 - HTFE should be labeled HIE to correspond to the rest of the manuscript.

Line 93 - polarity reversal of HIEs/HIOs produces apical-out, thus...

Line 94 - don't think it is possible to write that the central cavity is collapsed, there is no data for it. In Co et al., it is shown that the apical-out organoids still have a cavity and this paper does not show sufficient data to establish that in case of HIEs or HIOs there is no cavity.

Line 112 - authors claim that villin-specific stain is used to confirm polarity reversal. However, the Fig S1B description states that the stain is for vimentin. Originally, Figure S1A was villin, but that was replaced by MUC2 staining. Authors would need to amend the claim that polarity reversal was confirmed.

Figure 2D - the legend lacks description about what color is which marker. Assuming green is for villin and red/magenta for GBS, it seems that villin is potently stained inside the organoid and in the solution?, not just on the outer surface. Is this an artifact of the rendering? The Figure legend should specify which marker is which color.

Line 170 - Caco-2 cells instead of CaCO2.

Major comments:

Main focus of this study was whether apical-out HIEs or HIOs could serve as a representative model for GBS infection of pre-term infants. However, the data shown is not sufficient to support the conclusions of the study. The only data shown to describe the polarity reversal and impact on different cell types and tissue structures lacks important information e.g. no brightfield image or Hoechst staining for the markers in Figure S1.

Furthermore, authors claim they observed changes in the transcriptome upon GBS infection but only couple of genes were studied. The expression levels were normalized to different housekeeping genes for organoids and for a cell line making it hard to draw direct comparisons. The appropriate control for the organoids described here would be expression data for actual tissue samples, not from a cell line, to ensure that organoids model the situation well.

Number of papers appear to be cited in a confusing way. E.g. on line 164 the citation 20 does not describe comparison of expression with Caco-2 cells or demonstrate it as a suitable control for *in vivo* expression data. It also does not suggest that the genes *CLDN2*, *NOS2*, *OLFM4* are the ones with key differences between GBS infected and uninfected mice. The authors don't show transcriptomic analysis they claim to have done for HIE, HIO and Caco-2 and therefore it is not clear what they base their selection of the genes shown on. Line 113 – the reference 5 does not show cell marker staining for apical-out organoids since those are not used in the cited publication at all. Reference 5 describes using fetal-tissue derived organoids for microinjections – without polarity reversal. For HIO, images in Figure S1 are missing DAPI staining and/or brightfield image and the reader cannot evaluate whether the polarity reversal occurred the way authors claim it did. On line 169 – references 14 and 21 don't describe differences of organoids with Caco-2 and T-84 cell lines.

Figure S1 in general: The figure is missing nuclear staining or a brightfield picture to accompany the marker staining. This does not require any additional sample preparation that would impact the organoids, yet, it is not included. Therefore, the reader cannot assess the staining of the markers at all. This is especially true for Figure S1C, but for applies to all of them. Subfigure E then compares expression levels of some markers between a cell line Caco-2 and iPSC derived organoids, however, I don't think the expectation would be that they are the same. No data are shown for the HIE. The conclusion that there are "changes to the transcriptome" is not justified in my opinion. If anything, 6 specific genes are shown, 4 of them with a significant change in expression in HIO over Caco-2, but that is not representative of a transcriptome-wide shift and it is unclear if one or the other models the situation in preterm neonatal intestine.

What is the reasoning to use HIOs for modelling immature preterm neonatal intestine and its interaction with GBS?

Figure 5A - TEER measurement – this experiment lacks positive control for induced monolayer permeability.

Additional minor comments:

Figure 1 – HTFE should be labeled HIE to correspond to the rest of the manuscript.

Line 93 – polarity reversal of HIEs/HIOs produces apical-out, thus...

Line 94 – don't think it is possible to write that the central cavity is collapsed, there is no data for it. In Co et al., it is shown that the apical-out organoids still have a cavity and this paper does not show sufficient data to establish that in case of HIEs or HIOs there is no cavity.

Line 112 – authors claim that villin-specific stain is used to confirm polarity reversal. However, the Fig S1B description states that the stain is for vimentin. Originally, Figure S1A was villin, but that was replaced by MUC2 staining. Authors would need to amend the claim that polarity reversal was confirmed.

Figure 2D – the legend lacks description about what color is which marker. Assuming green is for villin and red/magenta for GBS, it seems that villin is potentially stained inside the organoid and in the solution?, not just on the outer surface. Is this an artifact of the rendering? The Figure legend should specify which marker is which color.

Line 170 – Caco-2 cells instead of CaCO₂.

The authors have addressed all my questions, and the paper can now be accepted for publication.

Reviewer #1:

Main comment:

1] I would recommend asking the authors to show high quality confocal microscopy data to establish whether the HIE can undergo polarity reversal and mimic the preterm intestinal tissue (actin, nuclei stained at minimum to confirm apical-out polarity). Authors should also amend their statements about transcriptomics and add important controls e.g. positive control for induced permeability in the TEER experiment for organoid-derived monolayers.

We thank the reviewer for this insightful comment. We performed actin and nuclei staining in addition to villin staining to parallel the polarity-reversal characterization shown in Co et al. The manuscript has now been updated to include these images (supplementary Figure S1) and explained in the results section:

“Apical-out orientation of HIOs and HIEs was confirmed by using a villin-specific stain to identify the epithelial brush border due to villin being a protein that is only expressed on the apical side of the intestinal epithelium as well as staining for F-actin along with nuclei (**Figure S1**). Dampened signaling of villin within basolateral-out HIE compared to the prominent signaling of villin in apical-out HIE further confirmed polarity reversal was achieved (**Figure S1A-B**). F-actin and nuclei staining of basolateral-out HIE revealed F-actin internal to the nuclei (**Figure S1C**) whereas in apical-out HIE F-actin revealed distinct prominence externally to the nuclei (**Figure S1D**). Staining of villin with nuclei was completed of both basolateral-out HIO (**Figure S1E**) and apical-out HIO (**Figure S1F**) verifying polarity reversal was accomplished similarly to HIE. F-actin and nuclei staining was also completed of basolateral-out and apical-out HIOs (**Figure S1G-H**). iPSC-derived HIOs contain both epithelial and mesenchymal cell populations, resulting in increased structural complexity. Because F-actin is expressed in multiple cell types, including mesenchymal cells, actin staining in HIOs does not selectively mark the apical brush border and therefore cannot reliably indicate polarity orientation. For actin staining to accurately indicate polarity in HIOs, the mesenchymal cells must be removed (as was done in Co et al. paper in figure 2), effectively generating a model that includes only epithelial cells like our enteroids.”

We have also amended our statements regarding transcriptomics to avoid implying that transcriptome-wide analyses were performed in this study; we now refer specifically to GBS-induced gene expression changes.

Finally, we have added the recommended positive controls to the TEER permeability assay. EGTA (2 mM) was included to demonstrate barrier disruption without significant

cytotoxicity, and Triton X-100 was used as a control for permeability loss due to cell death. These controls are now included in Figures 5A and 5B, and the manuscript text has been updated accordingly. We included a representative confocal image of the HIE infected monolayer (Figure 5D).

Major comments:

2] Main focus of this study was whether apical-out HIEs or HIOs could serve as a representative model for GBS infection of pre-term infants. However, the data shown is not sufficient to support the conclusions of the study. The only data shown to describe the polarity reversal and impact on different cell types and tissue structures lacks important information e.g. no brightfield image or Hoechst staining for the markers in Figure S1.

We thank the reviewer for this comment. Please note that previous Figure S1 is now Figure S3 in the revised manuscript. The imaging in Figure S3 was intended to validate the presence of cell types within the iPSC-derived human intestinal organoids (HIO), rather than to demonstrate polarity reversal. As noted previously, we acknowledge the limitations of these images; therefore, we removed these images and focused on qRT-PCR analysis to confirm presence of cell types in iPSC-derived HIOs (Figure S3).

Apical-out orientation is demonstrated in both model systems through F-actin and villin staining in fetal tissue-derived HIEs and iPSC-derived HIOs (Figure S1). Villin is a well-established apical brush-border protein, and prior work (Co et al., 2019) explicitly highlights its exclusive apical localization in intestinal epithelial cells. In our images, the DAPI-labeled nuclei are consistently positioned internal to the villin-positive apical surface in apical-out HIE/HIO, providing clear morphological confirmation of apical-out polarity in both HIEs/HIOs. Results section has been updated as noted in the main comment above.

3] Furthermore, authors claim they observed changes in the transcriptome upon GBS infection but only couple of genes were studied. The expression levels were normalized to different housekeeping genes for organoids and for a cell line making it hard to draw direct comparisons. The appropriate control for the organoids described here would be expression data for actual tissue samples, not from a cell line, to ensure that organoids model the situation well.

We thank the reviewer for this comment. Our intention was not to claim a transcriptome-wide analysis but rather to assess targeted gene expression changes

relevant to GBS pathogenicity. To contextualize these findings, we compared our qRT-PCR data with previously generated RNA-seq profiles from neonatal mouse intestinal tissue (reference 20), providing an *in vivo* benchmark for early-life intestinal responses. Additionally, the two organotypic systems used in this study—HIOs derived from induced pluripotent stem cells and HIEs derived from human fetal immature intestinal tissue—were selected specifically to approximate the immature human intestinal environment as closely as possible. These models therefore serve as biologically relevant comparators for evaluating GBS-induced responses in preterm-like intestinal epithelium. We also included Caco-2 cells because they remain a widely used reference model for studying GBS interactions within the intestinal epithelium. Although housekeeping genes differed between model types, our goal was not to make direct quantitative comparisons across systems but rather to highlight qualitative differences in GBS-induced gene expression patterns. In response to the reviewer's concern, we have revised the abstract to avoid implying transcriptome-wide analysis. The updated statement now reads:

“Using these models, we demonstrated that GBS induces changes in gene expression of both HIEs and HIOs that are distinct from what has been reported in immortalized adult cell lines.”

4] Number of papers appear to be cited in a confusing way. E.g. on line 164 the citation 20 does not describe comparison of expression with Caco-2 cells or demonstrate it as a suitable control for in vivo expression data. It also does not suggest that the genes CLDN2, NOS2, OLFM4 are the ones with key differences between GBS infected and uninfected mice. The authors don't show transcriptomic analysis they claim to have done for HIE, HIO and Caco-2 and therefore it is not clear what they base their selection of the genes shown on.

We thank the reviewer for this helpful clarification. Citation 20 was referenced to indicate that the genes highlighted in our study were identified as differentially expressed in our previously published RNA-seq datasets. Specifically, supplementary figure 2 of citation 20 includes validation of these genes in Caco-2 cells following GBS exposure, and supplementary figure 3 presents the selected genes in the RNA-seq analysis of neonatal mouse intestinal tissue. Our study does not present a transcriptome-wide analysis for HIEs and HIOs and as mentioned in the previous comment, we have revised the text to clarify this.

5] Line 113 - the reference 5 does not show cell marker staining for apical-out organoids since those are not used in the cited publication at all. Reference 5 describes using fetal-tissue derived organoids for microinjections - without polarity reversal.

We thank the reviewer for this comment. Reference 5 indeed describes fetal tissue–derived enteroids in the conventional basolateral-out orientation and does not include apical-out cultures. Our intention was to cite this work as evidence of validated epithelial cell-type composition in the basolateral state of these fetal-derived enteroids. The polarity-reversal methodology and characterization of apical-out enteroids are detailed extensively in reference 6, which demonstrates that apical-out cultures retain the same intestinal epithelial cell (IEC) lineages present in their basolateral-out counterparts. Based on these established findings, and our prior confirmation of IEC lineage composition in the basolateral orientation (reference 5), we were confident that the IEC lineage would be preserved following polarity reversal. Nonetheless, to provide direct validation within the context of the current study, we performed qRT-PCR for key epithelial cell-type markers, now presented in supplementary figure 2.

6] For HIO, images in Figure S1 are missing DAPI staining and/or brightfield image and the reader cannot evaluate whether the polarity reversal occurred the way authors claim it did.

We thank the reviewer for this comment. Please note that previous Figure S1 is now Figure S3 in the revised manuscript. As noted in our response to comment 2, the imaging in Figure S3 was designed specifically to validate epithelial cell-type presence in the iPSC-derived HIOs, not to demonstrate polarity reversal. We acknowledge that these images lack DAPI and brightfield channels; for this reason, we replaced the analysis with qRT-PCR to confirm the presence of key epithelial lineages in the HIOs (Figure S3). Assessment of polarity reversal in HIOs is instead shown in supplementary Figure S1, which provides clear morphological evidence of apical-out orientation. We have clarified this distinction in the revised text to avoid any confusion regarding the purpose of Figure S3.

7] On line 169 - references 14 and 21 don't describe differences of organoids with Caco-2 and T-84 cell lines.

We thank the reviewer for this helpful comment. We have revised the citation on line 169 to reference studies that directly compare organoid-based intestinal models (HIOs/HIEs) with Caco-2 and T84 cell lines. The updated references for this statement are now 13, 16, and 29, each of which appropriately addresses these differences. We have also re-evaluated the surrounding text to ensure that all citations throughout the manuscript are accurately placed following multiple rounds of revision.

8] Figure S1 in general: The figure is missing nuclear staining or a brightfield picture to

accompany the marker staining. This does not require any additional sample preparation that would impact the organoids, yet, it is not included. Therefore, the reader cannot assess the staining of the markers at all. This is especially true for Figure S1C, but for applies to all of them. Subfigure E then compares expression levels of some markers between a cell line Caco-2 and iPSC derived organoids, however, I don't think the expectation would be that they are the same. No data are shown for the HIE. The conclusion that there are "changes to the transcriptome" is not justified in my opinion. If anything, 6 specific genes are shown, 4 of them with a significant change in expression in HIO over Caco-2, but that is not representative of a transcriptome-wide shift and it is unclear if one or the other models the situation in preterm neonatal intestine. What is the reasoning to use HIOs for modelling immature preterm neonatal intestine and its interaction with GBS?

We thank the reviewer for the opportunity to clarify these points. Please note that previous Figure S1 is now Figure S3 in the revised manuscript. As noted in our responses to comments 2 and 6, the purpose of Figure S3 was to validate epithelial cell-type composition in the iPSC-derived HIOs, not to demonstrate polarity reversal. We acknowledge that the images lack nuclear staining and brightfield channels, and we agree that these additions would improve visualization. To address this limitation, we replaced the figure with qRT-PCR analysis to confirm the presence of key epithelial cell types in the HIOs (Figure S3E). We agree with the reviewer that expression levels between HIOs and Caco-2 cells would not be expected to match. This comparison was included to illustrate that HIOs contain multiple epithelial and supporting cell types that are absent in Caco-2 cells. While Caco-2 cells express markers such as CDX2 (enterocytes) and LGR5 (intestinal stem cells), they do not differentiate into other major intestinal lineages, including enteroendocrine cells (CHGA), goblet cells (MUC2), Paneth cells (LYZ), or mesenchymal cells (VIM). Our intention was therefore not to imply transcriptome-wide changes, but rather to validate the presence of diverse cell types using targeted qRT-PCR. To address the reviewer's concern regarding HIEs, we have now included parallel validation of epithelial cell-type markers in HIEs in supplementary Figure 2. Because the preterm intestinal epithelium contains multiple specialized cell types, we believe that models incorporating this cellular diversity such as HIOs and HIEs, more closely approximate the biology of the preterm neonatal intestine than Caco-2 cells, which are composed primarily of enterocyte-like cells and are nonetheless one of the most commonly used models for studying GBS interactions with the neonatal gut.

9] Figure 5A - TEER measurement - this experiment lacks positive control for induced monolayer permeability.

We thank the reviewer for this comment. In response, we have incorporated appropriate positive controls for induced monolayer permeability. Specifically, 2 mM EGTA was added to demonstrate barrier disruption without significant cytotoxicity, and Triton X-100 was included as a control for permeability loss due to cell death. These controls are now presented in Figures 5A and 5B, and the corresponding text in the manuscript has been updated in the results section accordingly as seen below:

“HIEs can be manipulated to form polarized monolayers with intact tight junctions between IECs as reflected by increasing TEER measurements over time. Upon GBS challenge, we noted a trend in increased monolayer permeability as compared to unexposed HIEs as indicated by a change in TEER. We included EGTA, a known disruptor of adherens and tight junctions, and Triton x-100 as controls. Similar to GBS, EGTA induced a trend in decreased permeability, while Triton x-100 treatment led to complete disruption of barrier function (**Figure 5A**). A lactate dehydrogenase (LDH) cytotoxicity assay was performed on media collected from each of the wells. Both GBS and EGTA treatments were not associated with IEC cytotoxicity, while Triton x-100 treatment induced cell death (**Figure 5B**). Recovery of GBS from media in the basolateral chamber indicates bacterial translocation through the epithelial monolayer (**Figure 5C**). These combined data suggests that translocation of GBS across HIE monolayers occurs without inducing cell death via either transcellular or paracellular pathways. A representative confocal microscopy image demonstrated GBS adherence to HIE monolayer following infection (**Figure 5D**).”

Additional minor comments:

1] Figure 1 - HTFE should be labeled HIE to correspond to the rest of the manuscript.

Resolved in figure.

2] Line 93 - polarity reversal of HIEs/HIOs produces apical-out, thus...

Line 94 - don't think it is possible to write that the central cavity is collapsed, there is no data for it. In Co et al., it is shown that the apical-out organoids still have a cavity and this paper does not show sufficient data to establish that in case of HIEs or HIOs there is no cavity.

We thank the reviewer for this comment. We agree that Co et al. do not explicitly discuss the presence or absence of a central cavity in apical-out organoids. In their Figure 1C–E, basolateral-out organoids display a clearly defined central lumen highlighted by apical ZO-1 localization, whereas apical-out organoids show apical markers redistributed to the outer surface. However, because Co et al. do not directly

address lumen morphology in the apical-out state, we recognize that our original phrasing may have implied a conclusion not fully supported by published data.

To avoid overinterpretation, we have revised the text to remove the statement that the central cavity is “collapsed” and now describe only the established and well-supported feature of polarity reversal, namely, the relocation of apical markers to the external surface of HIE/HIO.

3] Line 112 - authors claim that villin-specific stain is used to confirm polarity reversal. However, the Fig S1B description states that the stain is for vimentin. Originally, Figure S1A was villin, but that was replaced by MUC2 staining. Authors would need to amend the claim that polarity reversal was confirmed.

We thank the reviewer for this comment. Please note that previous Figure S1 is now Figure S3 in the revised manuscript. We agree that Figure S3 no longer contains villin staining, as the original villin panel was replaced during figure revision and moved to Figure 3H. As noted in our responses to major comments 2 and 6, villin is an apically localized protein routinely used to confirm apical-out orientation in HIEs and HIOs. In our study, polarity reversal is demonstrated in supplementary Figure S1. The manuscript has been updated to ensure that the claim regarding polarity confirmation is aligned with the correct figure.

4] Figure 2D - the legend lacks description about what color is which marker. Assuming green is for villin and red/magenta for GBS, it seems that villin is potentially stained inside the organoid and in the solution?, not just on the outer surface. Is this an artifact of the rendering? The Figure legend should specify which marker is which color.

We thank the reviewer for this comment. The colors corresponding to each marker are included at the end of the Figure 2 legend, where we specify: “[blue = nuclei (DAPI); green = villin (anti-villin IgG Alexa Fluor™ 488); magenta = GBS (anti-GBS IgG Alexa Fluor™ 568)].” To improve clarity, we have now moved this information earlier in the legend so it is more immediately visible to the reader.

5] Line 170 - Caco-2 cells instead of CaCO2.

Resolved in manuscript.

Reviewer #1:

Major Comments:

1] While the use of fetal tissue-derived organoids or organoids derived from iPSCs is interesting, I am lacking important information as well as controls. The study needs to include controls such as GBS-uninfected organoids etc. Perhaps they were performed but are not shown here, and the manuscript would benefit from a direct comparison.

We appreciate your interest in our organoid models. We have added representative images of uninfected HIE in Figure 2G and HIO in Figure 3G. Uninfected controls are included for all qPCR experiments as well.

2] It is not clear what the 3D visualization based on the confocal image adds to the data. Why is this type of visualization used? How was it rendered? The experimental information lacks sufficient details and the low quality of the original microscopy images leads to strange artefacts including merging nuclei of neighboring cells into one (e.g. Figure 2). Since microscopy data are a key portion of the paper, this is vital information.

We thank the reviewer for this thoughtful comment. We have clarified both the rationale and the technical details of the 3D visualization. The purpose of the Imaris renderings is to illustrate the spatial organization of cells and extracellular components within the organoid, which cannot be fully appreciated in 2D images or single optical planes. The 3D reconstruction provides volumetric context—highlighting the arrangement of peripheral cell layers surrounding a hollow lumen characteristic of mature organoids.

The 3D renderings were generated from confocal z-stacks collected at 0.5 μm intervals using a Zeiss LSM880 microscope (63 \times oil objective, NA 1.4). Surfaces were created in Imaris (v10.2, Oxford Instruments) using a fixed intensity threshold for each fluorescence channel (blue = nuclei, DAPI; green = villin for the epithelial border, anti-villin IgG Alexa Fluor™ 488; magenta = GBS, anti-GBS IgG Alexa Fluor™ 568). No manual smoothing or interpolation was applied beyond the default Gaussian filter (0.2 μm). The renderings are used solely for visualization of spatial relationships, not for quantitative analysis.

Regarding the appearance of merged nuclei, this effect results from the projection plane of the static image rather than from segmentation or rendering artefacts. Because the figure represents a single 3D orientation, overlapping nuclei along the z-axis can appear continuous in projection. We have added a supplementary rotational video (Figure 2D QR code in lower right quadrant) that demonstrates the full 3D structure of HIE, clearly showing discrete nuclei surrounding a central hollow core. We have also revised the figure legend and methods section to explain the

rendering parameters and imaging geometry in greater detail. Please note that previous figure 2 is now figure 3 in the revised manuscript and legend revised as noted below.

Figure 3. GBS adherence to apical-out epithelial border of human intestinal organoids.

Representative images using laser confocal scanning microscopy at (A) 63x magnification, (B) 1.5x zoomed in at 63x magnification, thus image is at 94.5x magnification (C) 100x magnification. Three-dimensional rendering of organoid architecture from confocal z-stacks images acquired using a Zeiss LSM880 microscope equipped with a 63× oil objective). Optical sections were collected at 0.5 μm intervals and rendered in Imaris (v10.2, Oxford Instruments) using the *Surface* tool with fixed intensity thresholds applied to each fluorescence channel [SD1] (D) staining of GBS [magenta = anti-GBS IgG Alexa Fluor 568], (E) nuclei [Blue = DAPI], (F) Composite image. (G) Uninfected HIO control image included for reference alongside GBS-infected HIO. Scale bars: Panel A, 20 μm; Panel B, 10 μm; Panel C, 5 μm; Panel D-F, 25 μm; Panel G, 10 μm.

3] In Figure S1, we lack a clear comparison of staining by different markers for the same organoid. Without a brightfield image or at least a nuclear staining image, it is impossible to assess the images, especially for panel D but also B. Ideally, we would look at a representative image of one organoid stained by multiple markers and then at a merged image.

We thank the reviewer for this helpful comment. We wish to clarify that the images in Figure S1 were direct visualizations of the raw confocal volumes. After reviewing the data, we determined that rendering these images would not improve clarity and could instead introduce artifacts or smoothing effects that obscure true staining patterns. Therefore, the images were retained in their raw form, processed only with the default Gaussian filter (0.2 μm) to minimize background noise while preserving signal integrity.

The iPSC-derived organoids shown in Figure S1 are structurally fragile, which limits the number of markers that can be reliably stained in a single section without compromising antigenicity. To represent key aspects of organoid organization, we selected the most robust stains across serial sections and included nuclear (DAPI) channels where available to aid orientation and interpretation.

To independently confirm cell-type composition, we have also performed PCR analysis of lineage-specific markers corresponding to the antigens visualized by immunostaining (Figure S1 E). We have updated the figure legend and methods section to clarify the imaging approach and the use of Gaussian filtering, and to emphasize that these panels represent distinct sections from iPSC-derived organoids.

4] GBS invasion into organoids and cells - since authors want to make conclusions about the

invasion of GBS, we need to see clearer data. For the intraorganoid invasion, we would need to see a confocal image going through an organoid cavity. 1D might look like the GBS is inside, but it is hard to make a conclusion since the section might be going through a surface layer. There is a lot of noise from villin antibody outside of the organoid itself, that the visualization in 1D removes, thereby "improving" the actually observed staining. Video in 1F only includes a part of an organoid, so we are not sure if the organoid was otherwise intact. For the intracellular localization - the visualization in 2D,F does not necessarily show intracellular localization and as described in my previous question - it is a 3D rendered visualization that appears to have artifacts.

We thank the reviewer for this detailed feedback and the opportunity to clarify the imaging approach. The images shown in the previously labeled Figure 1D–F (now Figure 2B–D) are indeed 3D renderings generated from complete confocal z-stacks using Imaris (v10.2, Oxford Instruments). The purpose of this rendering is to illustrate the spatial localization of *Group B Streptococcus* (GBS) within the three-dimensional architecture of the organoid — information that cannot be fully appreciated from single optical sections.

To minimize artefacts, surfaces were generated using the *Surface* tool with fixed, channel-specific intensity thresholds and the default Gaussian filter (0.2 μm) to reduce background noise. No manual interpolation or smoothing was applied. The underlying confocal stacks span the entire organoid, including the central lumen, and confirm that GBS signal extends into internal regions rather than being restricted to the surface. The static image in Figure 1D (now Figure 2B) represents one orientation within the full 3D volume, which can give the impression of surface localization when viewed in isolation.

We have now included an additional supplementary video showing the full rotation and z-projection of the same organoid (Figure 2D lower right quadrant), demonstrating both the intact structure and bacterial presence within the internal cavity and epithelial cell layer. Apparent background signal from villin outside the organoid arises from detached cell fragments and luminal debris typical of infected samples, not from rendering artefacts.

We have revised the figure legend and Methods section to explicitly describe the rendering parameters, the inclusion of the full z-stack volume, and the use of Gaussian filtering. Together, these clarifications provide transparency regarding data processing and demonstrate that the 3D visualization faithfully represents bacterial localization within intact organoids.

5] I think the publication would benefit from a brief explanation as to why did the expression data include these particular genes? How do these changes compare to other literature on transcriptomic changes after GBS infection e.g. <https://doi.org/10.1128/iai.00035-23> ?

We thank the reviewer for the opportunity to expand on this topic. Previous studies by our group using both a mouse model of GBS intestinal colonization as well as Caco-2 cell lines revealed

GBS-induced alterations in IEC gene expression — specifically in those related to barrier function. We conducted a transcriptomic analysis of the mouse model that revealed changes in the KEGG pathway responsible for epithelial barrier structure and function and validated those findings with Caco-2 cells through targeted qRT-PCR. This study prioritized the selected genes based on importance to epithelial barrier function upon GBS exposure across the models of Caco-2 cells, HIEs, and HIOs. We have updated the discussion section to include this explanation.

6] The organoids shown in microscopy don't appear to have a cavity. Is this representative of all organoids observed in the study?

We thank the reviewer for this observation. The images presented in the main figures are apical-out HIE/HIO that do not have a central cavity/lumen after completing the polarity reversal process. This morphology is representative of all apical out HIE/HIO in this study, however, during standard growth conditions of HIE/HIO before polarity reversal, the apical surface is faced inward creating the central cavity/lumen. As described in the manuscript, in this configuration access to the apical surface is limited, making apical out configured HIE/HIO ideal for visualizing direct host-pathogen interactions. The process of generating these apical out HIE/HIO, collapses the central lumen, causing the HIE/HIO to become spheroid-like in structure with the apical surface exposed externally. To clarify this process, the following was revised in the introduction: “The luminal surface of enterocytes, which is enclosed within the interior of HIE and HIO spheroids, is difficult to access within the central cavity. To overcome this challenge, polarity reversal of HIEs/HIOs is generated to produce apical-out HIEs and HIOs, thus collapsing the central cavity and allowing for the luminal apical surface of IECs to be in direct contact with reagents and/or microbes added to culture media [6].”

Minor Comments:

1] Labeling is not consistent throughout the paper.

Organoids are labeled as HIE/HIO, but then Figure 3 switches to S.I. Enteroids and Organoids. Caco-2 cells are labeled as CaCO₂ (chemical compound?) in multiple spots including Figure 3, but also in the text of the paper.

GBS is labeled as COH1 in Figure 4. I understand that that is the particular strain used, but for the sake of clarity, I would recommend using GBS as is used in the Figure 4 legend as well in the other cases where GBS is mentioned (e.g. other figures).

Resolved in figures and figure legend.

2] Scale bars - in Figure 1, the scale bars do not have a legible description, and it is not clear what they correspond to.

We thank the reviewer for this observation. Note that original Figure 1 is now Figure 2. Scale bars indicate the magnification and in Figure 2, represent 3 μm for Panel A, 10 μm for Panels B-F, and Panel G, 15 μm . Description has been added to the figure legend for both figures 2 and 3.

3] Is 1A a zoomed in image?

Figure 1A (now Figure 2A) is a zoomed in image as indicated by the scale bars. Figure 2A has a scale bar of 3 μm and the other panels in figure 2 have a scale bar of 10 μm . Figure 2 description has been updated as seen below:

Figure 2. Laser confocal scanning microscopy imaging of GBS-exposed HIE. (A) Increased magnification of GBS adhering to epithelial cell surface. **(B)** GBS is visualized within the core of apical-out enteroid. **(C)** 3D rendering of image D. **(D)** Two QR codes link to 3D Vimeo videos: The upper-left video displays nuclei and GBS rendered with raw filter of Villin, allowing enhanced visualization of internalized GBS while the lower-right video includes all components rendered—nuclei, GBS, and Villin. **(E-F)** GBS adhesion and invasion of HIE enteroid using 3D rendering of confocal images. **(G)** Representative image of HIE control (no GBS exposure) shown for contextual comparison. [blue = nuclei, DAPI; green = villin, anti-villin IgG Alexa Fluor™ 488; magenta = GBS, anti-GBS IgG Alexa Fluor™ 568] Scale bars: Panel A, 3 μm ; Panel B-F, 10 μm ; Panel G, 15 μm .

4] Image 2C seems to be compressed from the sides.

We thank the reviewer for bringing this to our attention. Note original Figure 2C is now Figure 3C. Image has been corrected.

5] Labeling in Figure 3 is incorrect. The legend description does not correspond to the plots shown in A, B and C. Perhaps the plots were shuffled around and the legend should be adjusted accordingly.

We thank the reviewer for finding this error. Note original Figure 3 is now Figure 4. Figures and figure legends have been updated.

6] Row 176 - in vivo should be italicized

Resolved in manuscript.

7] Human colonoid medium - please describe the actual medium. First two references Dame et

al. and Miyoshi and Stappenbeck do not describe "human colonoid medium". It is not clear what medium was used. I was not able to find a Michigan Protocols human colonoid medium.

We thank the reviewer for this comment and the opportunity to clarify. The full composition of the human colonoid medium (HCM) is described in detail in Reference 32, which includes a direct link to the Michigan Protocols. Within that resource, item #5 (“Complete gastrointestinal growth media [HCM...]*”) links to a Google document authored by Dame et al. that provides a comprehensive breakdown of the media components and preparation steps.

As noted in the manuscript, the Michigan protocol was adapted and further optimized using additional published work (References 34, 36-38). For example, Reference 37 (Miyoshi and Stappenbeck) describes the generation and formulation of LWRN-conditioned media, which comprises 65% of the HCM formulation. While we follow the Michigan protocol for HCM media, we also wish to acknowledge the foundational work from which that protocol was derived, as these sources informed the optimization of HCM for organotypic intestinal models.

8] The sentence on line 240 duplicates previous sentence, was the basement matrix Matrigel? Please, merge the two sentences to include all information for the sake of clarity. The basement matrix mentioned on line 246 is also Matrigel?

We thank the reviewer for this observation. All basement membrane matrix used for HIO was Matrigel. Sentence has been updated in the manuscript and now reads “16 μ L of basement membrane (Matrigel hESC –Qualified Matrix (Corning) diluted to a concentration of 2%) was combined with 1 mL of cold DMEM/F12 and 1 mL of the mixture was used to coat each well of the 6 well plate.

9] Line 281 - what factors were added to the medium? Only Y-27632 or others as well? What was the source of Y-27632?

We thank the reviewer for allowing us the opportunity to clarify line 281. The medium used for Day 10, stage 3 organoid formation was Intestinal Organoid Growth Medium by combining STEMdiff™ Intestinal Organoid Basal Medium and STEMdiff™ Intestinal Organoid Supplement as described in the Stem Cell Technologies protocol. No other additives were included. Methods have been updated to clarify.

10] Polarity reversal protocol, line 310 - how many days after passaging was the polarity reversal performed? Could the authors include the images on polarity reversal comparison between day 0 and 3 to essentially compare unreversed vs reversed organoids? This would be of value since the polarity reversal appears to be slower compared to adult enteroid shown in Co et al.

We thank the reviewer for the valuable feedback. Polarity reversal was initiated 5 to 7 days after passaging, once the HIE had re-established structural integrity. While we appreciate the suggestion to include comparative images from days 0 to 3, we respectfully note that the HIE are highly fragile during the early stages of reversal and do not tolerate fixation and staining well at that time. As such, imaging at these time points was not feasible. Polarity reversal was confirmed through staining for villin localized to the apical surface, as shown in the representative images provided. These images reflect the successful establishment of apical-out polarity and serve as the basis for validating the reversal.

11] The schemes in Figure 5 are useful, but could perhaps be moved in the manuscript to Figure 1 to explain the workflow in the manuscript as sort of a graphical summary?

We appreciate the suggestion to reposition the schematic in Figure 5 as a graphical summary earlier in the manuscript and has now been moved to **Figure 1**. The following sentence was added to the Results section: A schematic in **Figure 1A** illustrates the generation of HIE/HIO and **Figure 1B** depicts the subsequent immunostaining procedure.

12] I would avoid the use of the term spheroids, since those are different from organoids.

We thank the reviewer for this comment and the opportunity to clarify why this term is used in this context. In this manuscript, the term “spheroids” is used specifically to describe the early-stage aggregates prior to full organoid maturation. This nomenclature aligns with the terminology provided by Stemcell Technologies, which refers to these structures as spheroids during the initial phase of differentiation. At this stage, the cells have not yet developed the complex architecture or functional features characteristic of mature organoids. We have used this term intentionally to reflect the developmental progression described in the manufacturer’s protocol and to distinguish between pre-organoid and organoid stages. The Methods section has been updated to convey this description.

13] Line 367 - Dapi should be DAPI.

Resolved in manuscript.

14] The expression data should include a GBS-untreated control and a detailed explanation about normalizations done in the methods section. Currently, it is not clear how the data was processed before being plotted.

We thank the reviewer for the opportunity to clarify the GBS-untreated control. Targeted qRT-PCR was performed using TaqMan™ Gene Expression Assays (Thermo Fisher Scientific). Glyceraldehyde 3-phosphate dehydrogenase (GAPDH) was used as the reference gene for both

HIE and HIO samples, and β -actin was used for Caco-2 cells. Each cell type was analyzed under two conditions: untreated (GBS-untreated control) and GBS-exposed. HIEs included four biological replicates per condition with three technical replicates, while HIOs and Caco-2 cells included three biological replicates per condition with three technical replicates. The Ct values of the target genes were normalized to their respective housekeeping genes, and relative quantification was performed using the $\Delta\Delta$ -CT method. The untreated control condition was used as the baseline reference, with its expression level normalized to 1.0 for \log_2 fold change calculations. All plotted data represent the \log_2 fold changes in expression of each target gene relative to this baseline. The Methods section and figure legend have been revised to reflect this explanation.

Reviewer #2:

1] Did authors test the co-cultures of HIEs/HIOs with other innate immune cells (e.g., macrophages, neutrophils)?

We thank the reviewer for their shared interest in co-culture models with HIE/HIO. We have not yet performed co-culture experiments involving HIEs/HIOs with innate immune cells such as macrophages or neutrophils. However, we are actively developing a model within our laboratory to investigate the interactions between HIEs/HIOs and macrophages in the context of Group B Streptococcus (GBS) exposure, with the goal of elucidating epithelial-immune dynamics during GBS adhesion, invasion, translocation and permeability.

2] What is the cell death mechanism (apoptosis, necrosis, or other forms of cell death) in intestinal epithelium upon exposure to GBS?

In this work we did not characterize cell death mechanisms triggered by GBS challenge in this manuscript. The literature would suggest that GBS-induced epithelial death is likely mostly programmed cell death (apoptosis, pyroptosis, or necroptosis) at physiologic infection levels but may also be mediated by direct lysis when exposed to high concentrations of the GBS pore forming toxin β -Hemolysin/Cytolysin. We measured cell viability in our HIE monolayers and noted it was greater than 80% cell viability following GBS exposure for 2 hours.

3] What are the major gene networks that are upregulated or downregulated in response to GBS in the organoid models, and how do these differ from those in immortalized cell lines?

We appreciate the reviewer's interest in the transcriptional responses to GBS in intestinal organoid models. However, this manuscript is focused on methodological development and does not include transcriptomic analyses. RNA-seq was not performed as part of this study, and characterization of gene networks upregulated or downregulated in response to GBS exposure is beyond the scope of the current work. Observed differences between selected genes of interest

suggest differential response that is dependent on model type (transformed, HIE, HIO). Future studies incorporating RNA-seq will be essential to elucidate these differential gene expression profiles and further define the molecular mechanisms underlying host-pathogen interactions in physiologically relevant models.

Re: Spectrum02316-25R2 (Bridging the Gap: Organotypic Models to Study Late-onset Group B Streptococcus Infection)

Dear Dr. Alexia N Pearah:

Your manuscript has been accepted, and I am forwarding it to the ASM production staff for publication. Your paper will first be checked to make sure all elements meet the technical requirements. ASM staff will contact you if anything needs to be revised before copyediting and production can begin. Otherwise, you will be notified when your proofs are ready to be viewed.

Sincerely,
Nagendran Tharmalingam
Editor
Microbiology Spectrum